# The ice-nucleating ability of quartz immersed in water and its atmospheric importance compared to K-feldspar

Alexander D. Harrison[1], Katherine Lever[1], Alberto Sanchez-Marroquin[1], Mark A. Holden[1,2*], Thomas F. Whale[1,2], Mark D. Tarn[1,3], James B. McQuaid[1] and Benjamin J. Murray[1]

[1]School of Earth and Environment, University of Leeds, Leeds, LS2 9JT, UK
[2]School of Chemistry, University of Leeds, Leeds, LS2 9JT, UK
[3]School of Physics, University of Leeds, Leeds, LS2 9JT, UK
* Now at School of Physical Sciences and Computing, University of Central Lancashire, Preston PR1 2HE, UK

**Abstract.** Mineral dust particles are thought to be an important type of ice-nucleating particle (INP) in the mixed-phase cloud regime around the globe. While K-feldspar has been identified as being a particularly important component of mineral dust for ice nucleation, it has been shown that quartz is also relatively ice nucleation active. Given quartz typically makes up a substantial proportion of atmospheric desert dust it could potentially be important for cloud glaciation. Here, we survey the ice-nucleating ability of 10 α-quartz samples (the most common quartz polymorph) when immersed in microlitre supercooled water droplets. Despite all samples being α-quartz, the temperature at which they induce freezing varies by around 12°C for a constant active site density. We find that some quartz samples are very sensitive to ageing in both aqueous suspension and air, resulting in a loss of ice-nucleating activity, while other samples are insensitive to exposure to air and water over many months. For example, the ice nucleation temperatures for one quartz sample shifted down by ~2°C in 1 hour and 12°C after 16 months in water. The sensitivity to water and air is perhaps surprising as quartz is thought of as a chemically resistant mineral, but this observation suggests that the active sites responsible for nucleation are less stable than the bulk of the mineral. We find that the quartz group of minerals are generally less active than K-feldspars by roughly 7 °C, although the most active quartz samples are of a similar activity to some K-feldspars with an active site density, $n_s(T)$, of 1 cm$^{-2}$ at -9 °C. We also find that the freshly milled quartz samples are generally more active by roughly 5 °C than the plagioclase feldspar group of minerals and the albite end-member has an intermediate activity. Using both the new and literature data, active site density parameterisations have been proposed for freshly milled quartz, K-feldspar, plagioclase and albite. Combining these parameterisations with the typical atmospheric abundance of each mineral supports previous work that suggests that K-feldspar is the most important ice-nucleating mineral in airborne mineral dust.

## 1 Introduction

The formation of ice in supercooled clouds strongly affects hydrometeor size which in turn impacts cloud lifetime, precipitation and radiative properties (Kanji et al., 2017). There are a number of primary and secondary mechanisms through which ice can form in clouds. Homogeneous freezing of cloud droplets becomes increasingly important below -33 °C (Herbert et al., 2015), but clouds commonly glaciate at much warmer temperatures (Kanitz et al., 2011;Ansmann et al., 2009). Freezing at these warmer temperatures can occur through secondary ice production (Field et al., 2017) or heterogeneous freezing on ice-nucleating particles (INPs) (Murray et al., 2012;Hoose and Möhler, 2012). The presence of INPs, which tend to comprise only a small fraction of cloud condensation nuclei, can dramatically reduce the lifetime of shallow clouds (Vergara-Temprado et al., 2018), and alter the development of deep convective clouds through, for example, the release of latent heat which invigorates the updraft thus altering cloud structure (Lohmann, 2017;Rosenfeld et al., 2011). It is also recognised that an accurate representation of cloud phase is important for assessments of climate sensitivity (Tan et al., 2016;Ceppi et al., 2017). However, our understanding of which type of aerosol particles serve as effective INPs is incomplete (Vergara-Temprado et al., 2017;Kanji et al., 2017).

Mineral dust has been inferred to be an effective INP in the atmosphere from field, model and laboratory studies (Hoose and Möhler, 2012;Vergara-Temprado et al., 2017). Observations of aerosol within ice crystals have shown that mineral dust is often present, suggesting they act as INPs within mixed phase clouds (Iwata and Matsuki, 2018;Eriksen Hammer et al., 2018;Pratt et al., 2009). Laboratory studies also demonstrate mineral dusts are relatively effective at nucleating ice (Hoose and Möhler, 2012;Murray et al., 2012;DeMott et al., 2015). Atmospheric mineral dusts are composed of several components. Clay is a major component of airborne mineral dust and is sufficiently small that its atmospheric lifetime is relatively long. Hence, historically ice nucleation studies have focused on the clay group of minerals (e.g. Broadley et al., 2012;Murray et al., 2011;Wex et al., 2014;Mason and Maybank, 1958;Pinti et al., 2012;Roberts and Hallett, 1968;Hoose and Möhler, 2012). However, more recent work shows that K-rich feldspars (K-feldspars) are very effective INPs when immersed in supercooled water (Whale et al., 2017;Zolles et al., 2015;Tarn et al., 2018;Peckhaus et al., 2016;DeMott et al., 2018;Reicher et al., 2018;Harrison et al., 2016;Niedermeier et al., 2015;Atkinson et al., 2013). However, there are other minerals present in the atmosphere, many of which are relatively poorly characterised in terms of their ice-nucleating activity.

Quartz is a major component of aerosolised atmospheric mineral dust (Perlwitz et al., 2015;Glaccum and Prospero, 1980) and studies have shown that it can be active as an INP (Zolles et al., 2015;Atkinson et al., 2013;Isono and Ikebe, 1960;Holden et al., 2019;Kumar et al., 2019a;Losey et al., 2018). Boose et al. (2016) showed a correlation between the INP activity of nine desert dusts and the concentration of K-feldspar at temperatures of -20°C. However, at lower temperatures (-35 to -28 °C) the ice-nucleating activity of the dusts correlated with the combined concentration of quartz and K-feldspar. Boose et al. (2016) thus emphasised the importance of understanding quartz and feldspars present in the atmosphere for the modelling of INPs. Recently, Kumar et al. (2018) investigated five milled quartz samples (two synthetic, three naturally occurring) for their ice-nucleating activity, demonstrating the activity of milled quartz. Very recently, Holden et al. (2019) demonstrated that nucleation on quartz is indeed site specific, through repeated freezing experiments with high-speed cryomicroscopy, and found that micron sized defects tended to be collocated with the nucleation sites. While our understanding of ice nucleation by quartz has improved recently, it is still unclear how variable quartz samples are in their ice-nucleating ability, which prevents an assessment of its atmospheric importance as an ice-nucleating particle relative to other minerals.

When designing experiments focused on understanding the ice-nucleating activity of atmospheric mineral dusts, we must consider the processes that lead to the production of dust in the atmosphere (these processes are illustrated in Figure 1). It is common practice to mill relatively pure samples to fine powders which can be studied in the laboratory (Atkinson et al., 2013;Harrison et al., 2016;Peckhaus et al., 2016;Zolles et al., 2015;DeMott et al., 2018;Kumar et al., 2018;Niedermeier et al., 2015) for the purposes of characterising the ice-nucleating ability of individual minerals, but the relevance of this mechanical milling process to natural airborne mineral dusts needs some discussion. Ultimately, atmospheric mineral dust is derived from bulk rocks which are mechanically broken down to finer particles through erosion processes (Blatt et al., 1980). The finer material that results can be transported by rivers or wind and forms soils in deserts or fertile regions. The particles in these soils undergo complex ageing chemistry (and biology), which converts certain minerals into clay minerals (Wilson, 2004). Minerals such as pyroxenes and amphiboles are relatively readily converted to clays over geological timescales,

but quartz and to a lesser extent feldspars are relatively inert and therefore persist in soils (Goldich, 1938;Wilson,
2004). However, the ageing state of the surfaces of these minerals is unclear. While ageing processes may modify
the surfaces relative to the original fresh surfaces, these aged materials are continually exposed to aeolian
processes that involve grains mechanically abrading against one another, resulting in rounding of grains and the
break-up of aggregates (Bagnold, 1941;Pye, 1994). These vigorous aeolian processes result in the generation of
small airborne dust particles which most likely have fresh surfaces. Hence, the commonly applied practice of
mechanically milling rock samples for laboratory characterisation has some justification, but it would be wise to
test how sensitive the active sites on these surfaces are to exposure to air and water. Previous studies indicate that
K-feldspars tend to be relatively insensitive to exposure to water and air (Harrison 2016; Whale 2017), although
acids can deactivate K-feldspars (Kumar et al., 2018). Hence, in the absence of strong acids, freshly milled K-
feldspar is thought to be relevant for atmospheric mineral dust. Quartz on the other hand has been shown to be
very sensitive to exposure to water and re-milling these samples appears to readily expose or create new active
sites (Zolles et al., 2015;Kumar et al., 2019a).
In this study we present a survey of the ice-nucleating ability of 10 naturally occurring quartz samples and
demonstrate the variability in ice-nucleating ability within natural quartz. We also explore the stability of a subset
of these samples to time spent in water or exposed to air confirming that the activity of some quartz samples are
very sensitive to ageing, in contrast to K-feldspars. Then, in order to compare the potential contribution of quartz
to the atmospheric INP population to that of other minerals we have generated a parameterisation for freshly
milled quartz based on the experimental work in this study. In addition we present new parameterisations for K-
feldspar, plagioclase feldspar, and albite feldspar based on datasets available in the literature. This allows us to
compare the potential contribution of quartz, albite, plagioclase and K-feldspar to the atmospheric INP population.
**2 Quartz, the mineral**
Quartz is the second most abundant mineral in the Earth's crust after the feldspar group of minerals. Its hardness
(Moh's scale 7) and chemical nature along with its lack of cleavage planes mean it is also a common constituent
of sands and soils as it is resistant to weathering processes. Although quartz does not have cleavage planes it does
exhibit conchoidal fracturing meaning particles tend to have smoothly curving surfaces as a result of fracturing
(Deer et al., 1966), rather than planes with steps that might be expected on a cleavage plane. As it is a common
constituent to soils, including desert soils, it can be lofted into the atmosphere and is found within transported
mineral dusts (Caquineau et al., 1998;Avila et al., 1997;Kandler et al., 2011;Kandler et al., 2009).
The silica minerals are composed of $SiO_2$ tetrahedra with each silicon being bonded to four oxygen atoms and
these tetrahedra form a 3D framework which can be in six or eight membered loops (Deer et al., 1992). There are
three principle crystalline types of $SiO_2$: quartz, cristobalite and tridymite, with stishovite and coesite being other
high pressure polymorphs. The polymorph that is present depends on the temperature and pressure during
formation (Koike et al., 2013;Swamy et al., 1994). All three crystalline silica types (quartz, cristobalite and
tridymite) can exist in two polymorphs, both a high temperature (β) and low temperature (α) state. α-quartz is
most commonly found at or near the Earth's surface due to it being the most stable at atmospheric conditions and
thus is the dominant polymorph of quartz found in soils and in atmospheric desert dust aerosol (Deer et al., 1992).
In fact, α-quartz is so common that by convention it is referred to simply as quartz.
Generally, quartz samples tend to be close to 100 % $SiO_2$ although it is common to find small amounts of oxides
as inclusions or liquid infillings within cavities (Deer et al., 1966). The substitution of $Al^{3+}$ for $Si^{4+}$ allows for the
introduction of alkali ions such as $Li^+$ and $Na^+$. These subtle impurities can lead to a variety of colours. If quartz
with impurities (for example Al) is exposed to low levels of naturally occurring radiation then one pair of electrons
from an oxygen adjacent to Al can be emitted leaving unpaired electrons otherwise known as "hole defects"
(Nassau, 1978). This forms the basis for colour centres, which cause the colouration of amethyst. Amethyst is
typically violet in colour and differs from standard α-quartz in that it has a larger proportion of $Fe_2O_3$ inclusions
and marginally more $TiO_2$ and $Al_2O_3$ in its structure (Deer et al., 1966). Rose quartz generally contains higher
amounts of alkali oxides, $Fe_2O_3$, $TiO_2$ and $MnO_2$ (Deer et al., 1966). It has a pinkish colour which is thought to be
attributed to the presence of a fibrous mineral which was first suggested to be dumortierite (Kibar et al.,
2007;Applin and Hicks, 1987) but has been suggested to be a different, unclassified type of mineral (Goreva et
al., 2001). Smoky quartz has a black colour which is caused by colour centres created by the irradiation of iron
(Nassau, 1978). Chalcedony is a form of cryptocrystalline or microcrystalline α-quartz (Deer et al., 1966). It has
been suggested that it is also commonly intergrown with another polymorph of quartz known as moganite (Heaney
and Post, 1992;Götze et al., 1998). Moganite has a monoclinic crystal structure opposed to the trigonal crystal
system of quartz. Chalcedony often includes micropores within its structure due to its microcrystalline nature
(Deer et al., 1966).

**3 Materials and Methods**
**3.1 Samples and preparation**
10 α-quartz samples were tested for their ice-nucleating ability. These included four typical α-quartzes, two
amethysts, two microcrystalline quartzes (chalcedony), one rose quartz and one smoky quartz, as summarised in
Table 1. Photographs of the samples are presented in Figure 2. These samples were selected to investigate the
natural variability of the ice-nucleating ability of α-quartz.
These samples were sourced from various gem sellers. The minerals were visually inspected, using their colour,
crystal habit, lustre and cleavage to confirm whether the mineral was quartz and, if so, what type of quartz.
Rietveld refinement of powder X-ray diffraction (XRD) patterns was then used to verify the silica polymorph and
identify any significant crystalline impurities. The results of this process are presented in Table 1. Raman
spectroscopy was used in conjunction with XRD to test for the presence of moganite within the two chalcedony
samples based on the work of Götze et al. (1998). However, both methods indicated that no moganite was present
above the limit of detection (~1 wt%).
Eight of the samples were prepared from bulk rock or crystal samples by first rinsing the rock surface with
isopropanol and pure water and placing in a clean sealed plastic bag before chipping off fragments and then
grinding them into a powder with an agate mortar and pestle. The mortar and pestle were cleaned before use by
scrubbing them with quartz sand (Fluka) and rinsing thoroughly with pure deionised water and isopropanol. A
similar method was employed by Harrison et al. (2016) who investigated less ice-active minerals (plagioclase
feldspars) and found that contamination from the cleaning process was not observed. Atkinson quartz (the same
quartz sample as used by Atkinson et al. (2013)) and Fluka quartz were supplied as a powder, although Atkinson
quartz was originally ground via the same milling process (Atkinson et al. 2013).
These samples were reground to ensure all samples initially had freshly exposed surfaces for ice nucleation
experiments. The milling process was used to break down mineral crystals/powders to a sufficient size so that
they may be suspended in water. We argue that these freshly milled samples are relevant in that they represent
the fresh surfaces which are likely produced by mechanical processes in nature as rocks are broken down and
particles aerosolised through the saltation process (see Figure 1 and discussion in the introduction). We
therefore suggest that the results from these freshly ground samples of quartz represent an upper limit to the ice-
nucleating ability of quartz in atmospheric mineral dust since ageing processes may reduce this activity.
The specific surface areas of the quartz samples were measured using the Brunauer-Emmett-Teller (BET) $N_2$
adsorption method with a Micromeritics TriStar 3000 instrument (Table 1). Heating of the sample at 100 °C
overnight was performed under a steady flow of dry nitrogen to evaporate any moisture in the sample before the
surface area measurement. After BET analysis, 1 wt% suspensions for all the samples were prepared
gravimetrically by suspending a known amount of material in purified water (18.2 MΩ cm at 25 °C) in a 10 mL
glass vial. In some instances we had small amounts of sample and so the sample used for BET analysis was
subsequently used for the succeeding ice-nucleation experiments. As quartz is a hard mineral the use of magnetic
stirrer bars was avoided when suspending the material as preliminary experiments showed the potential for the
Teflon coating to abrade off the stirrer bars and become mixed with the suspension. We also chose not to use glass
stirrer bars, partly because glass is softer than quartz and partly because we have noted in the past that is can be a
source of contamination. Therefore particles were suspended by vortexing for 5 mins prior to ice nucleation
experiments. Only small amounts of sample were available for Mexico quartz and Uruguay amethyst and so the
powder used for BET analysis was then used to prepare the suspensions for ice nucleation experiments. The BET
analysis and subsequent suspension in water was carried out within a week of grinding the sample.
**3.2 Ice nucleation experiments**
The microlitre Nucleation by Immersed Particle Instrument (μL-NIPI) was employed to test the ice-nucleating
ability of the various quartz samples in the immersion mode (Whale et al., 2015). This technique has been used
in several previous ice nucleation studies e.g. (Atkinson et al., 2013;O'Sullivan et al., 2014;Harrison et al., 2016).

In brief, 1 μL droplets of a suspension were pipetted onto a hydrophobic glass cover slip atop a cold plate (EF600, Asymptote, UK). During pipetting, the suspension was vigorously shaken manually every 10 droplets (with roughly 40 droplets per experiment) to keep the quartz particles suspended and to ensure that the amount of mineral in each droplet was similar. The cold plate and glass slide were then enclosed within a Perspex chamber and a digital camera was used to image the droplets. The temperature of the cold plate was decreased at a rate of 5 °C min$^{-1}$ to 0 °C (from room temperature), then at 1 °C min$^{-1}$ until all the droplets were frozen. Whilst cooling the system, a gentle flow of zero grade dry nitrogen (<0.2 L min$^{-1}$) was passed across the cold plate to reduce condensation onto the glass slide, which can cause interference between freezing droplets and the surrounding unfrozen droplets (Whale et al., 2015). As the droplets were cooled, images were recorded with the digital camera and freezing events identified in post analysis to calculate the fraction of droplets frozen as a function of temperature ± 0.4 °C. A second run for each sample suspension, with a fresh array of droplets, was performed immediately after the first experiment with approximately 1 hour between the two runs. Prior to the start of each experimental day droplets of pure water (no dust) were used to determine the background freezing signal. Umo et al. (2015) compiled a large collection of background freezing results to create a fit which represents the variability of the background in the μL-NIPI instrument. The background signal measured in this study was in line with the lower bound set by Umo et al. (2015).

We assume that nucleation on quartz occurs at specific active sites, as supported by the work of Holden et al. (2019) who showed that nucleation occurs preferentially at specific sites on α-quartz and feldspar using high-speed cryomicroscopy of ice crystal growth on thin sections of mineral. The cumulative ice-nucleating active site density $n_s(T)$, on cooling from 0 °C to a temperature, $T$, was determined for each quartz sample. Standardising the active site density to the surface area of nucleant allows for comparison of the ice-nucleating ability of different materials (Connolly et al., 2009; Vali et al., 2015). It should be noted that this model neglects the time dependence of nucleation, which can have some influence on the nucleation temperature (Herbert et al., 2014; Holden et al., 2019). $n_s(T)$ is calculated using:

$$\frac{n(T)}{N} = 1 - \exp(-n_s(T)A), \qquad (1)$$

where $n(T)$ is the cumulative number of frozen droplets on cooling, $N$ is the total number of droplets in the experiment. $A$ is the surface area of nucleant per droplet calculated based on the mass of quartz per droplet (assumed to be the same as in the bulk suspension) and the specific surface area determined via BET analysis.

We conducted Monte Carlo simulations to estimate the error in $n_s(T)$ as a result of the randomness of the distribution of active sites in the droplet freezing experiments. These simulations consider the possible distribution of active sites throughout the droplets that explain each fraction frozen and quantify this uncertainty, which is then combined with the uncertainty in the pipetting and BET measurements. This methodology was based on the work of Wright and Petters (2013).

**4 Results and discussion**

**4.1 The variable ice-nucleating ability of α-quartz**

The cumulative fraction of droplets frozen ($n(T)/N$) on cooling is shown in Fig. 3a for arrays of droplets containing the quartz samples. Comparison of these curves with the fraction frozen curves for droplets without added particles in the μL-NIPI system (Umo et al., 2015), shows that all quartz samples heterogeneously nucleate ice since the freezing temperatures for droplets containing quartz are always much higher than the pure water droplets. These fraction frozen curves are then translated into $n_s(T)$ in Fig. 3b-c. In Fig. 3b we show $n_s$ for freshly prepared samples where the particles were suspended in water for ~10 minutes before carrying out an experiment. The variability in the ice-nucleating ability of these α-quartz samples is striking. Bombay chalcedony and Atkinson quartz are substantially more active than the other samples with the activity spanning roughly 10 °C at $n_s(T) = 10$ cm$^{-2}$. While the overall spread is large, it is also notable that the droplet freezing temperatures of 8 out of 10 of the samples fall between -17 °C and -20 °C at $n_s(T) = 10$ cm$^{-2}$.

In Fig. 3c we show $n_s$ for both the first (fresh) run and a subsequent run performed approximately one hour after the first experiment for each quartz sample. In the cases of Bombay chalcedony, Brazil amethyst and Smokey quartz, the first and second runs where identical within the uncertainties, whereas in the other cases there was a systematic decrease in freezing temperature. For example, the temperature at which Atkinson quartz had an $n_s(T)$ of 1 cm$^{-2}$ decreased by ~3 °C between the first experiment and the second experiment run. In the past, using this

technique with mineral particles of a similar grain size has mostly resulted in consistent results from run-to-run
(e.g. Atkinson et al., 2013; Whale et al., 2015). This suggests that the decrease in activity seen for some quartz
samples is a real change in the activity of the quartz rather than artefacts such as, for example, the settling of
particles out of suspension leading to less surface area in each droplet. The finding that the activity of many of the
$\alpha$-quartz samples decrease with time spent in water is perhaps surprising given quartz is typically regarded as an
inert material. We come back to this issue of ageing of active sites in water and air in section 4.2 where we
describe a dedicated set of experiments to explore this issue.
The Bombay chalcedony sample stands out as being one of the most active quartz samples. For $n_s = 10$ cm$^{-2}$ the
Bombay chalcedony nucleates ice at -9 °C which is comparable to K-feldspar (see section 5.1, for a comparison
with other minerals). As described in section 2, chalcedony is a microcrystalline form of $\alpha$-quartz and commonly
has micropores. It is possible that these micropores contain ice nucleation active sites or create zones of weakness
which allow defects to be created when ground. In order to test if the superior ice-nucleating ability of Bombay
chalcedony is inherent to chalcedony, we located, characterised and tested a second chalcedony sample. Grape
chalcedony has a similar microcrystalline form to Bombay chalcedony, but behaves more like the other quartz
samples we have tested, both in having a lower ice-nucleating activity, but also in the decrease in its activity with
time spent in water. One possibility is that the Bombay chalcedony sample is contaminated with another very
active ice-nucleating component. The X-ray diffraction results suggest that there is not enough inorganic
crystalline material, for example K-feldspar, to account for the result. In addition, we washed a ~2 g sample of
unground Bombay chalcedony in 10 mL pure water (shaking vigorously for ~2 minutes) and tested the water. A
droplet freezing assay with this washing water indicated that there was no significant detachable contamination.
This suggests that the ice-nucleating activity of the Bombay chalcedony is inherent to the material rather than
associated with an impurity, although the presence of an ice-nucleating impurity cannot be categorically excluded.
These results suggests that a subtle difference between the two chalcedony samples causes the Bombay
chalcedony to be much more active.
The second most active quartz sample, fresh Atkinson quartz, does not have any obvious differences with the
other less active quartz samples which might explain its activity. It is almost entirely pure $\alpha$-quartz with only a
minor component of calcite (0.2%). It is unlikely that the calcite component is responsible for nucleation since
Uruguay amethyst contains the same percent impurity of calcite and is much less ice active.
Overall, the results in Fig. 3 show a surprising diversity in ice nucleation behaviour. As mentioned above, quartz
is a relatively uniform material which is chemically and physically stable, hence we might have expected its ice-
nucleating ability to be uniform and insensitive to ageing processes (in fact, this was our original hypothesis when
we started this project). However, the results clearly demonstrate neither of these expectations is correct. Since
all these quartz samples are $\alpha$-quartz we might have expected all of these quartz samples to exhibit similar
nucleating properties. This variability indicates that these quartz samples do not nucleate through a lattice
matching mechanism. This is consistent with the recent observation that nucleation on quartz occurs at active sites
(Holden et al., 2019). Our results suggest that these active sites have diverse properties, with different activities,
different site densities and some being sensitive to ageing processes where others are not. In the next section we
present a set of experiments designed to further probe the ageing of the ice nucleation sites on quartz samples.
**4.2 The sensitivity of ice-nucleating activity with time spent in water and air**
The results presented in Fig. 3 clearly indicate that the activity of many of the samples of quartz decreases by
several degrees within an hour (Fig. 3b). In initial experiments we also showed that the quartz powder used by
Atkinson et al. (2013) had lost its activity since it was initially tested. The sample had been stored in air within a
sealed glass vial under dark conditions for ~5 years. However, milling of the powder dramatically increased its
activity, which suggests that milling can (re)expose surfaces with the most effective active sites. This observation
is similar to that described by Zolles et al. (2015) who noted that two out of three quartz samples increased in
activity by up to 5 °C on milling. This supports the hypothesis that fresh surfaces are often key to maximising a
quartz sample's ice-nucleating ability. Very recently, Kumar et al. (2018) have also observed that milling quartz
increases its ice nucleation activity and suggest that this may be a result of defects created during the process.
In order to further explore the stability of active sites we tested how the activity of three samples of quartz varied
when exposed for a range of times to water and air. For this investigation we tested: i) Smoky quartz, as it is a
representative quartz in terms of its ice-nucleating ability, lying within the middle of the spread of $n_s(T)$; ii)
Bombay chalcedony, as it was the most active sample and iii) Atkinson quartz, since initial experiments indicated
it was highly sensitive to ageing in both water and air. The dry powder and suspension samples were stored at
room temperature in a dark cupboard in sealed glass vials. Prior to the droplet freezing experiment, wet samples
were agitated to re-suspend the particles and the dry powders were added to water in the standard manner
described above. The $n_s(T)$ of the various quartz samples aged in both water and in air for varying times are
displayed in Fig. 4.
Each of the three samples responded in a distinct manner to time spent in water. Inspection of Fig. 4 (a, c and e)
reveals that while the ice-nucleating ability of Smoky quartz did not significantly decrease after ~1 h, its activity
decreased by about 3 °C after four months in water which is well outside the uncertainties of the experiment.
Bombay chalcedony was far more stable in water, with no substantial change in the $n_s(T)$ curve after four months,
being within 1 °C of the fresh sample (close to the uncertainties of the experiment). In contrast, the activity of
Atkinson quartz decreased dramatically on exposure to water. Even after only ~1 hour in suspension the $n_s(T)$
curve decreased by 2 °C, but after 16 months in water the activity decreased by 12 °C. These results point to
populations of very different active sites on these three different quartz samples.
We also found that the activity of some quartz samples decreased even when they were stored in air (Fig. 4b, d
and e). Dry Smoky quartz and Bombay chalcedony powders were tested after being left in a glass vial for 20
months and showed no decrease in activity. In contrast the activity of Atkinson quartz decreased by ~5 °C in half
of this time period (10 months). Fig. 4f also shows the initial freezing temperatures obtained using the same
sample from the Atkinson *et al*. (2013) study which had been stored for ~5 years in a glass vial. This sample was
~10 °C less active compared to the freshly ground powder.

### 308 4.3 Discussion of the nature of active sites on quartz

These results paint a complex picture of the properties of the active sites on quartz samples. Not only is the
absolute activity of the samples variable, but the sensitivity of the sites to time spent in water and air is also highly
variable. The active sites of the Atkinson quartz are far more susceptible to ageing in water and air than both the
Smoky quartz and Bombay Chalcedony. The sites on Bombay chalcedony are stable in both air and water, whereas
those on Smokey quartz are somewhat intermediate in stability, being sensitive to water only after an extended
period of time beyond 1 hour.
Very recently, Kumar et al. (2018) also described the deactivation of quartz in suspension over a period of five
days. However, they noted that time series experiments carried out within glass vials showed deactivation of
quartz in pure water whereas experiments within polypropylene falcon tubes did not. They suggested that silicic
acid leached from the glass vial walls allows the quartz fragments to slowly grow and the active sites to be lost
during the process. The explanation of Kumar et al. (2018) is consistent with our observation that the nucleating
ability of many samples decreases with time spent in water. However, it is inconsistent with the stability of
Bombay chalcedony and it cannot explain the loss of activity seen for Atkinson quartz when aged in air.
The physical and chemical characteristics which lead to the large variability in the properties of the ice nucleation
sites on quartz are challenging to define. Classical nucleation theory suggests that ice critical clusters at the
nucleation temperatures observed in this study are likely to be on the order of several nanometres across (Pummer
et al., 2015). It therefore seems reasonable to think that the relevant ice nucleation sites will be on a similar scale
but the nature of these sites remains unclear. A molecular dynamics study by Pedevilla et al. (2017) suggested
that surfaces with strong substrate-water interaction and high densities of OH groups (or other H-bonding groups)
give rise to effective sites for ice nucleation. However, sites with high densities of surface OH groups are also
inherently thermodynamically unstable and will have a tendency to either react with, for example, moisture in air,
or rearrange to a more stable configuration. Hence, it may be at defects in the crystal structure where such sites
become stabilised when the thermodynamic cost of having a nanoscale region with a high density of H-bonding
groups is outweighed by the gain from relaxing strain in a structure. For example, in K-feldspar, it has been
suggested that active sites are related to strain induced by exsolution into K and Na rich regions, which is known
to result in an array of nanoscale topographical features (Whale et al., 2017). Consistent with this idea, Kiselev
et al. (2016) reported that nucleation on K-feldspar was related to exposed patches of high energy (100) and
Holden et al. (2019) demonstrated that nucleation on K-feldspar always occurs within micrometre scale surface
imperfections. Holden et al. (2019) reports that topographic features were observed on quartz, at some of the
nucleation sites, but they have not been further characterised.

Larger nanoscale patches of surface H-bonding groups should be better at nucleating ice, but these larger high energy patches will also be less energetically stable. Hence, one might expect that the sites responsible for nucleation at the highest temperatures would also be the least stable and most sensitive to time spent in water or air. But, this does not hold for Bombay chalcedony which is the most active quartz we studied and also the most insensitive to exposure to water and air. This indicates that the sites in this case are either of a completely different chemistry (perhaps a different high energy crystal plane), or the topography and strain associated with a defect imparts a greater stability on these sites. The fact that Bombay chalcedony is distinct from the bulk of the samples in being a microcrystalline quartz may be related to this, however, Grape chalcedony also has a similar morphology and does not possess the population of very active sites.

The increased ice nucleation associated with milling may be caused by the mechanical fracturing of the quartz leading to exposure of high energy but unstable sites, which decay away through a structural rearrangement process when exposed to air or liquid water. Alternatively, milling may simply result in the removal of reaction products to leave exposed active sites. Kumar et al. (2018) suggest the milling process causes the breakage of Si-O bonds which act as high energy sites for ice nucleation. Quartz does not exhibit a preferential plane of weakness (cleavage) to break along and it therefore fractures. The presence of small impurities distributed throughout the lattice, as described in sections 2 and 3, may influence the nature of fracturing and hence create differing defects and high energy sites. Gallagher (1987) classified impurities as a form of structural weakness. The impurities can create zones of weakness and stress within the crystal structure and therefore act as a pathway of least resistance resulting in the breakage of bonds and development of microtexture. Alternatively, in some instances the impurities may create areas of greater strength and so fracturing occurs around these zones. Hence, it is possible that the presence of impurities influences the way in which individual quartz samples fracture and therefore influence the presence of active sites.

Inherently, quartz is rather simple in terms of naturally occurring defects compared to other minerals, such as feldspar. In fact, in the past quartz has been considered to be in the 'perfect crystal class', i.e. lacking imperfections. However, quartz does have defects, albeit at a lower density than other minerals (Spencer and Smith (1966)). Quartz minerals can be subject to varying conditions and stresses after their formation and so the geological history of the quartz may also influence the degree of microtexture. For example, a quartz sample which has undergone stress at a fault boundary is more likely to exhibit microtextural features than one that has not (Mahaney et al., 2004). It may be these microtextural differences that lead to the observed variability ice-nucleating ability. This hypothesis might be tested in the future if quartz samples could be obtained with well characterised geological histories.

It has also been observed in the past that, for other minerals, the specifics of the mineral formation mechanism are critical for determining its ice-nucleating ability. Whale et al. (2017) demonstrated that a sample of K-feldspar, which had cooled sufficiently quickly during its formation that it did not undergo exsolution and therefore lacked the associated microtextures, had very poor ice nucleation properties. This was in contrast to the more common K-feldspars which do have exsolution microtexture and nucleate ice very effectively. Despite having very different ice-nucleating properties, their crystal structures and compositions are very similar. A similar formation pathway dependence may be true for quartz, such as strain introduced in geological fault systems. But one thing is clear: while bulk mineralogy is a guide to ice-nucleating activity, in some cases details of the formation pathway may be more important.

**5 The importance of quartz relative to feldspar for ice nucleation in the atmosphere**

**5.1 Comparison to the literature data for quartz and feldspar**

The data from the present study are contrasted with literature active site density data for quartz (Zolles et al., 2015;Atkinson et al., 2013;Losey et al., 2018) in Fig. 5. This data is also compared with $n_s(T)$ parameterisations for desert dust samples (Niemand et al., 2012;Ullrich et al., 2017) and K-feldspar (Atkinson et al., 2013). The variability within the α-quartz samples that we report is also reflected in the literature data for quartz. It is striking that two of the quartz samples in this study, Bombay chalcedony and Atkinson quartz, have an activity approaching or equal to K-feldspar. Nevertheless, it is apparent that quartz is never substantially more active than K-feldspar or desert dust in terms of $n_s(T)$.

Since one of our objectives is to determine how effective quartz is at nucleating ice in comparison to feldspars, we contrast the literature active site density data for feldspars and quartz in Fig. 6. The feldspars have been colour

coded into the plagioclase (blue), albite (green) and the K-feldspar (orange-reds) groups. We note that, by
convention, albite is considered part of the plagioclase solid solution series. However, Harrison et al. (2016)
demonstrated that albites had a distinct nucleating activity and therefore we plot them here as a separate group.
The K-feldspars presented here represent the K-rich samples from the alkali feldspar group (i.e. >10 % K). Overall
there is a general trend in that plagioclase feldspars are the least active of the four mineral groups and K-feldspar
is the most active. Both albite and quartz show similar, intermediate, activities. K-feldspars from Whale et al.
(2017) which did not exhibit the common phase separation were excluded from this plot as they are
unrepresentative of common K-feldspars and are rare in nature. Although quartz is an ice active mineral, Fig. 6
supports the consensus that it is the K-feldspars that are the most active mineral for ice nucleation that is commonly
found in mineral dusts in the atmosphere.

## 5.2 New parameterisations for the ice-nucleating activity of quartz, K-feldspar, plagioclase and albite

In order to be able to determine which mineral is most important in the atmosphere we need the activity of each
mineral (expressed as $n_s(T)$) in combination with estimates of the abundance of each mineral in the atmosphere.
In this section we produce new $n_s(T)$ parameterisations for quartz, K-feldspar, plagioclase and albite using data
from the present study in addition to literature data.
The new set of parameterisations are shown in Fig. 7. In order to derive these parameterisations we compiled
data for representative samples of quartz, K-feldspar, plagioclase and albite. To create these parameterisations we
binned the data within each dataset into 1 °C intervals and then fitted a polynomial line through the log averages
of the data. We binned the data in an attempt to remove bias towards datasets with relatively high data density. In
addition, we only applied a fit in the temperature range where multiple datasets were present (with the exception
of plagioclase, where the available data is so sparse in some temperature regimes that we had to relax this criterion
in order to produce a parameterisation). We used polynomial fits to represent the data since the data is quite
complex and alternatives such as a straight line would produce a very poor representation of the data. These fits
were constrained at the warmest and coldest temperatures in order to obtain a reasonable representation of the
data at these limits. We stress that these fits must not be extrapolated to higher and lower temperatures. The
standard deviation for each parameterisation was calculated by taking the average of the standard deviations of
the log $n_s(T)$ values for each 1 °C temperature interval. The corresponding value was then used to approximate the
standard deviation from each fit, which is represented by the dashed lines and shaded area in Fig. 7.
For the quartz fit, the chalcedony samples were excluded given these microcrystalline minerals are
unrepresentative of most quartz in nature and that they are therefore likely to be in negligible abundances in the
atmosphere. We also only include the runs with freshly made quartz suspensions in the parameterisation since the
second runs often showed signs of deactivation in suspension. By only using the relatively fresh suspension data,
our parameterisation is representative of freshly milled quartz dust. The new parameterisation can be seen in Fig.
7a-b and covers a temperature range of -10.5 °C to -37.5 °C and nine orders of magnitude in $n_s(T)$. This is the first
robust $n_s(T)$ parameterisation developed for this mineral that can be used to determine its role as an INP in the
atmosphere.
The K-feldspar parameterisation developed by Atkinson et al. (2013) has been used extensively within the ice
nucleation community. However, this parameterisation was created with data from one K-feldspar sample and
does not reflect the variability we now know to exist. The parameterisation developed as part of this study can be
seen in Fig. 7c-d. We excluded K-feldspar samples which did not exhibit phase separation from the Whale et al.
(2017) study from this parameterisation as these types of alkali feldspar are rare and unlikely to be found in
significant quantities in the atmosphere. The data included in these plots includes all three polymorphs of K-
feldspar (microcline, orthoclase and sanidine), although most of the data is for microcline. The strongly
hyperactive TUD #3, examined by Harrison et al. (2016) and Peckhaus et al. (2016), was excluded as it exhibited
extremely high activity and appears to be an exceptional case which is generally unrepresentative of the K-feldspar
group of minerals. With this in mind we have developed a parameterisation which represents K-feldspars that
possess exsolution microtexture. It should be noted that all of the studies used BET derived surface areas for the
calculation of $n_s(T)$ other than DeMott et al. (2018) and Augustin-Bauditz *et al.* (2014) who used geometric surface
areas. However, while the difference between BET and geometric surface areas is substantial for clay samples
(Hiranuma et al., 2015), the discrepancy is much smaller for materials with larger grain sizes like feldspar
(Atkinson et al. 2013). When the new K-feldspar parameterisation is compared to the literature data it represents
the variability of K-feldspar, as well as the curvature in the datasets. In particular, the new parameterisation
captures the observed plateau in $n_s(T)$ below about -30 °C. In addition, the new parameterisation produces higher
$n_s(T)$ values at temperatures warmer than -10 °C relative to that of Atkinson et al. (2013). Below -10 °C this new
parameterisation gives lower values of $n_s(T)$. The temperature range of the parameterisation is also extended,
covering -3.5 °C to -37.5 °C.
The parameterisation proposed here to represent plagioclase feldspar is shown in Fig. 7e-f. The parameterisation
spans a temperature range of -12.5 °C to -38.5 °C. Only one dataset was available to represent the plagioclase
feldspars in the lowest temperature regime (Zolles et al., 2015), hence this parameterisation needs to be used
cautiously, but it is nonetheless a best estimate at present given the current data available. A similar caution must
be accepted when using the albite parameterisation displayed in Fig. 6g-h which spans a range of -6.5 °C to -35.5
°C. For the albite parameterisation, the hyperactive Amelia albite from the Harrison et al. (2016) study was
excluded due to its exceptional ice-nucleating ability making it unrepresentative of the other five albite samples.
Hence, this parameterisation is representative of the non-hyperactive albites.
The parameterisations are summarised in Fig. 8a and are then combined with a typical abundance of each mineral
to estimate the INP concentration ([INP]$_T$) associated with each of the four minerals in Fig. 8b. On average,
roughly 3±6 % (by mass) of atmospheric transported mineral dust particles are K-feldspar whereas 16±15 % are
quartz and 8±3 % are plagioclase (see compilations of measurements in (Atkinson et al., 2013), which are
consistent with more recent measurements (Boose et al., 2016b)). Albite is often grouped with plagioclase
feldspars when determining the mineralogy of atmospheric mineral dusts rather than being reported on its own.
For the purposes of this estimate we have assumed that albite has a concentration equal to 10 % of that of
plagioclase. [INP]$_T$ was derived from the $n_s(T)$ parameterisations assuming a surface area concentration of mineral
dust of 50 $\mu m^2$ $cm^{-3}$ (a moderately dusty environment) and assuming that the mass fraction of each mineral is
equivalent to its surface area fraction. In order to approximate the size distribution of dust, a lognormal size
distribution centred around particles of 1 $\mu m$ in diameter with a standard deviation of 0.3 was used. We have also
assumed that each mineral is externally mixed (see Atkinson et al. (2013) for details of how to treat the mixing
state of mineral dust), which is the assumption that has been made when modelling the global distribution of INP
in the past (Atkinson et al., 2013 and Vergara-Temprado et al., 2017). In reality, desert dust aerosol will be
somewhat internally mixed. The opposing assumption of full internal mixing produces 1-2 orders more INP at the
lowest temperatures, but produces the same INP concentration above about -25°C (Atkinson et al. 2013). The
upper and lower bounds for each line in Fig 8b are derived from the range of mineral mass concentrations.
The [INP]$_T$ curves in Fig. 8b confirm that under most atmospheric situations K-feldspar has the main contribution
to the ice-nucleating particle population in desert dust. Quartz is the next most important mineral, with plagioclase
the least important. The contribution of pure albite is rather uncertain given the amount of pure albite in desert
dust is poorly constrained, but it is unlikely to compete with K-feldspar. Nevertheless, while K-feldspar is the
most important contributor to the INP population, the estimates in Fig. 8b do suggest that quartz may make a non-
negligible contribution to the INP budget at temperatures between about -20 and -12.5 °C. This is particularly so
when we consider the variability in the ice-nucleating ability of the K-feldspar and quartz groups. It is possible
that in a desert dust aerosol that if the K-feldspar was at the bottom end of the activity, whereas the quartz were
at the top end of its activity range, then the quartz would contribute more INP than K-feldspar. However, it should
also be considered that the estimated [INP]$_T$ curves in Fig. 8b are also based on the assumption that quartz has the
activity of fresh quartz. We know from the work presented above that the activity of quartz is sensitive to ageing
processes. We cannot quantify ageing of atmospheric quartz, but the parameterisation we present here probably
represents an upper limit to its activity. In contrast, the activity of K-feldspar does not decrease with time spent in
water or air (Kumar et al., 2018;Harrison et al., 2016;Whale et al., 2017). Overall, we conclude that K-feldspar
contributes the bulk of the INPs associated with desert dust, because it is more active and it is less sensitive to
ageing processes. However, we should not rule out quartz making a significant contribution to the INP population
in a minority of cases.
**5.3 Testing the new parameterisations against literature laboratory and field measurements of the ice-**
**nucleating ability of desert dust**
We now test the quartz and K-feldspar parameterisations to see if they are consistent with literature data of the
ice-nucleating ability of desert dust (Fig. 9). In Fig 9a we contrast the predicted $n_s(T)$ values, based on the quartz
and K-feldspar parameterisations, against a variety of literature datasets for desert dust. For the K-feldspar based
prediction, we have presented lines where 20 %, 1 % and 0.1 % of the surface area of dust is made up of K-
feldspar. For the 20 % prediction, which is consistent with measurements in Cape Verde (Kandler et al., 2011),
we have also shown the natural variability in K-feldspar activity as the shaded region. The line assuming quartz
is the dominant ice-nucleating mineral in desert dust is for 12 % quartz which again is consistent with
measurements made in Cape Verde (Kandler et al., 2011).
From Fig. 9a it is clear that quartz does not account for the $n_s(T)$ measurements of desert dusts sampled directly
from the atmosphere and suspended in laboratory studies. However, the new K-feldspar parameterisation is
consistent with the ice-nucleating activity of dusts over a wide range of temperatures. The K-feldspar
parameterisation reasonably represents the majority of mineral dust measurements when taking into account that
typically ~1 % to 25 % of atmospheric desert dust can be attributed to K-feldspar (Atkinson et al., 2013) and that
there is a natural variability in the ice-nucleating ability of K-feldspar (as presented by the shaded area around the
20 % K-feldspar prediction). The shape of the parameterisation represents the bulk of the data well and plateaus
at the lowest temperatures in agreement with the observations.
Fig. 9b shows INP concentrations measured from an aircraft in the eastern tropical Atlantic (Price et al., 2018)
plotted with the predicted INP concentrations based on the K-feldspar parameterisation developed by Atkinson et
al. (2013) (in black dashed lines), the parameterisation for desert dust by Niemand et al. (2012) (blue dashed lines)
and the K-feldspar parameterisation proposed here (red solid lines). The parameterisations were calculated
assuming an externally mixed scenario (although both internal and external mixing assumptions produce a similar
result in the regime where the measurements were made). The upper and lower bounds were calculated by
incorporating the maximum and minimum in the aerosol surface area concentrations corresponding to the various
aircraft measurements (23.8 $\mu m^2\ cm^{-3}$ to 1874 $\mu m^2\ cm^{-3}$) (Price et al, 2018). K-feldspar was assumed to represent
20 % of the aerosol surface area, based on measurements by Kandler et al. (2011). Note that the small number of
data points above ~-11°C have a very high uncertainty due to Poisson counting issues and should be regarded as
upper limits. Price et al. (2018) and Sanchez-Marroquin et al. (2019) have described a sub-isokinetic sampling
bias in the aircraft inlet which results in an enhancement of aerosol surface area by roughly a factor of 2.5 for the
used sampling conditions. We have therefore corrected the Price et al. (2018) data downwards by a factor of 2.5
(although on the log scale this makes a relatively small difference).
We can see that the Atkinson et al. (2013) parameterisation is a relatively poor predictor of the INP concentration,
especially at temperatures colder than about -15 °C. The parameterisation by Niemand et al. (2012) tends to over-
predict INP concentrations relative to the Price et al. (2018) data by about one order of magnitude. The K-feldspar
parameterisation proposed here better represents the magnitude, the range and the slope of the aircraft data.
Overall, the new K-feldspar parameterisation provides a good representation of the ice-nucleating activity of dust
from field and laboratory studies and it is also clear that quartz is of second order importance for desert dust's ice-
nucleating ability.

## 2    Conclusions

We have studied 10 quartz samples for their ice-nucleating ability in order to better understand and define the ice-
activity of this abundant mineral. The chosen samples were all α-quartz, the most common silica polymorph
found at the Earth's surface, but included a variety of α-quartz types with varying degrees of impurities and
different crystal habits. We found that the ice-nucleating activity of these samples is surprisingly variable,
spanning about 10 °C. Eight out of ten of the quartz samples lay within -17 °C to -20 °C at $n_s(T) = 10\ cm^{-2}$, with
two quartz samples, Bombay chalcedony and Atkinson quartz, being much more active (as active as K-feldspar).
Overall, the quartz group of minerals tend to be less active than the K-feldspars, slightly less active than albite,
but more active than the plagioclase feldspars. In the future it would be interesting to probe the nature of the active
sites on the two most active samples and to try to contrast these sites to those on the less active samples in order
to further understand the nature of active sites and why they have such strongly contrasting characteristics.
Although quartz is regarded as a relatively chemically inert mineral the activity of some samples decreases with
time spent in air and water. Most of the samples were sensitive to time spent in water, but interestingly, the most
active sample's activity did not change significantly even after many months in water. We note that the sensitivity
to time in water displayed by most of the quartz samples studied here is in strong contrast to K-feldspars, which
tend to be much more stable. Related to this, we also note that solutes can alter the ice-nucleating ability of mineral
samples (Whale et al., 2018;Kumar et al., 2018;Kumar et al., 2019a, b). Sensitivity to these ageing processes and
solutes could be very important in determining the dominant INP types globally (Boose et al., 2019). Hence, we
suggest further studies aim to build a better understanding of the relationship between the experimental
observations and field collected samples to determine the role of ageing in the atmosphere.
To investigate the relative importance of quartz to feldspars in the atmosphere we have proposed new active site
density parameterisations for quartz, K-feldspar, plagioclase and albite. These parameterisations are based on a
combination of the data presented here for quartz along with data available in the literature. Sparse data sets
available for the albite and plagioclase mineral groups lead to lower confidence when creating parameterisations
for these mineral groups. It is suggested that future studies expand on the current datasets of the ice-nucleating
behaviour of minerals to improve these parameterisations. When using the newly developed parameterisations to
predict INP concentrations in combination with typical atmospheric abundances of minerals, it is found that K-
feldspar typically produces more INP than milled quartz (or any other mineral). Also note that the parameterisation
for quartz is for freshly milled quartz and the ageing results presented here and elsewhere (Zolles et al.,
2015;Kumar et al., 2019a) suggest that the active sites on quartz are removed on exposure to air and water.
Therefore the parameterisation for milled quartz should be regarded as an upper limit. Even with this upper limit,
quartz is of secondary importance relative to K-feldspars which appear to be less sensitive to ageing processes. In
addition, we find that the newly developed K-feldspar parameterisation is consistent with $n_s(T)$ literature
measurements on desert dusts and better represents field measurements of INP concentrations in the dusty tropical
Atlantic compared to the parameterisations by Atkinson et al. (2013) and Niemand et al. (2012). We hereby
propose the use of this new K-feldspar parameterisation when predicting INP concentrations related to mineral
dusts.


*Data availability.* Data for the various quartz samples presented in this paper are available at
http://dx.doi.org/10.5285/171726739bb54d0ba84cdde15c5b17ae.
*Author contributions.* ADH designed the experiments with help from scientific discussions with BJM, TFW and
JBM. Both KL and ADH performed the experiments. AS completed the calculations for the external mixing
assumption used in figures 8b and 9a and assisted in the calculation of errors for the active site density
measurements. MAH carried out Raman analysis of the chalcedony samples and MDT helped in the assembly of
the literature data. ADH prepared the manuscript with contributions from all co-authors.
*Acknowledgements.* We would like to take the opportunity to thank Lesley Neve for her contribution in the XRD
analysis. The authors acknowledge the European Research Council (ERC, MarineIce: 648661), Engineering and
Physical Sciences Research Council (EPSRC, EP/M003027/1) and the Natural Environment Research Council
(NERC, NE/M010473/1) along with Asymptote Ltd. (now part of GE Healthcare) for funding this research.

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

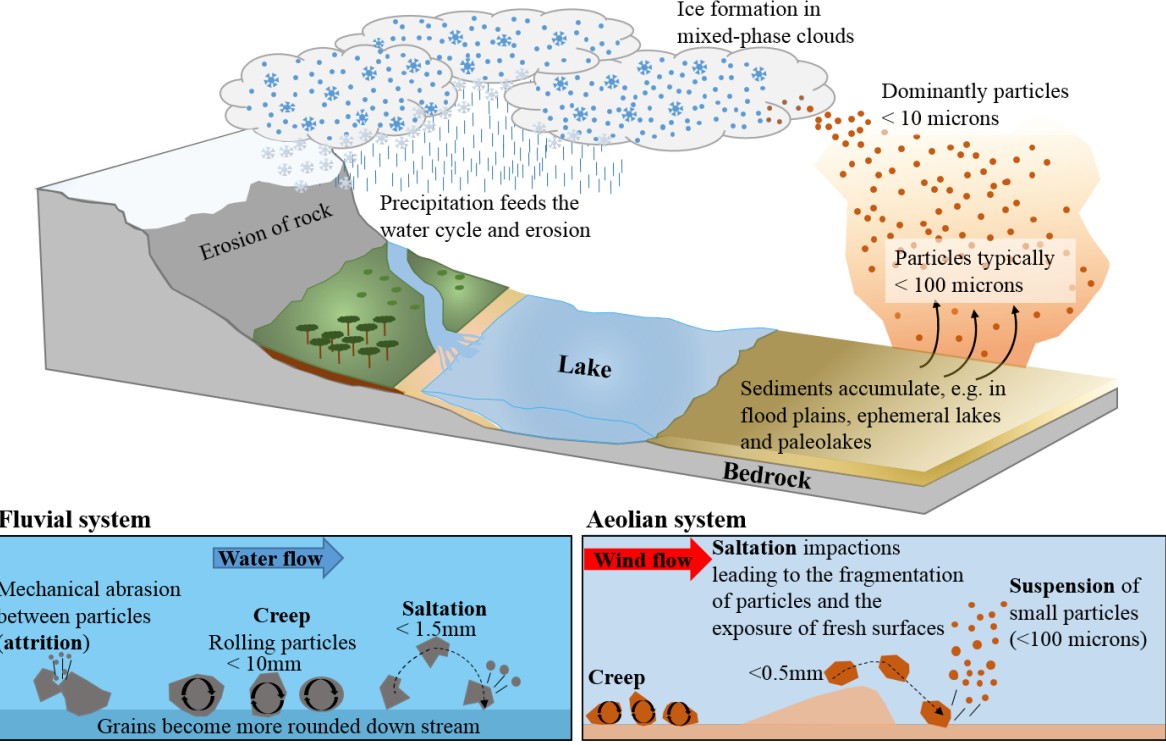


**Figure 1:** Illustration of the geological processes that lead to the creation of a mineral dust and its emission to the
atmosphere. Saltation and creep occur both in fluvial and aeolian systems and are processes responsible for the movement
of material. There is typically more energy in a fluvial system and hence the particle sizes moved in these systems are
larger. Attrition between particles causes them to fragment and to become more rounded on transport with small particles
becoming suspended in water or air. The particles smaller than ~10 µm in air can be transported for long distances and
may interact with clouds, serving as INP, many 100s or 1000s of kilometres away from the source regions.


| Sample | XRD analysis | BET surface area $(m^2g^{-1})$ | Spherical equivalent diameter (μm) |
|---|---|---|---|
| Bombay chalcedony | α-quartz: 100% | 1.23 ± 0.01 | 1.88 |
| Grape chalcedony | α-quartz: 100% | 4.39 ± 0.01 | 0.53 |
| Smoky quartz | α-quartz: 98.3%<br>Haematite: 0.1%<br>Albite: 1.6% | 1.23 ± 0.01 | 1.88 |
| Rose quartz | α-quartz: 100% | 1.13 ± 0.01 | 2.04 |
| Atkinson quartz | α-quartz: 99.9%<br>Calcite: 0.1% | 4.20 ± 0.01 | 0.55 |
| Fluka quartz | α-quartz: 100% | 0.91 ± 0.01 | 2.54 |
| Mexico quartz | α-quartz: 96.4%<br>Dolomite: 3.6% | 1.74 ± 0.01 | 1.33 |
| LD1 quartz | α-quartz: 100% | 0.94 ± 0.01 | 2.45 |
| Uruguay amethyst | α-quartz: 99.9%<br>Calcite: 0.1% | 1.46 ± 0.01 | 1.58 |
| Brazil amethyst | α-quartz: 100% | 2.76 ± 0.01 | 0.84 |


**Table 1:** Table showing the relative concentrations of different minerals within each sample and the respective BET
specific surface area of the ground sample and derived spherical equivalent average surface area. The uncertainty in the
XRD analysis is on the order of 0.1 %, hence the identification of some trace constituents in some samples is tentative.


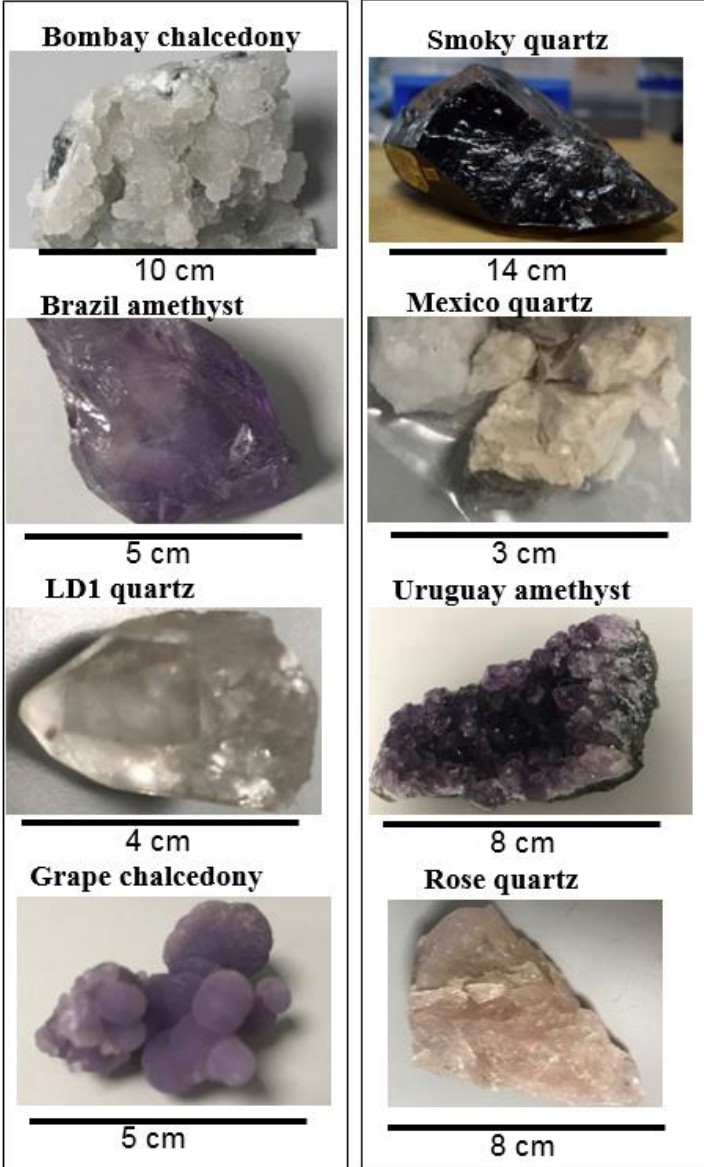



**Figure 2:** Pictures of the various quartz samples explored in this study showing their varying appearances and characteristics.
Samples supplied in a milled state are not shown.








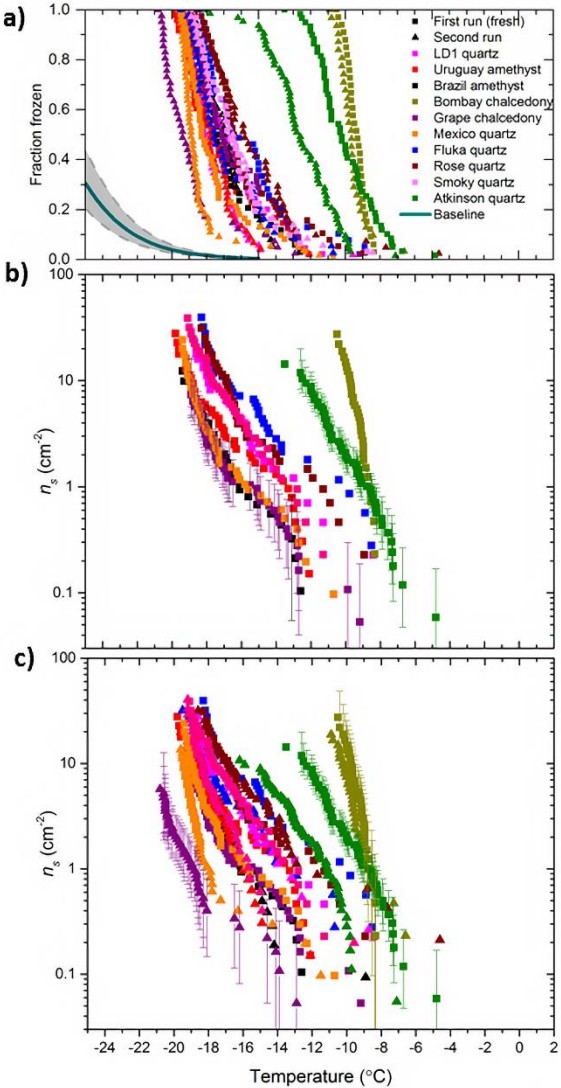



**Figure 3:** Fraction frozen and active site densities for ten quartz suspensions (1 wt%). **a)** The fraction frozen versus temperature for the different quartz samples investigated in this study. The range of freezing for the baseline is highlighted in the grey shaded region (Umo et al., 2015). **(b)** The active site density ($n_s$) for the range of quartz samples in this study. In this plot only the first run of each sample is displayed. These samples are considered to be fresh as they have only spent roughly 10 minutes in suspension. **(c)** The active site density ($n_s$) versus temperature for the quartz samples on their initial runs and their corresponding second runs. The second runs were carried out roughly an hour after the first run. A sample of the error bars are shown in Fig. 2b/c.




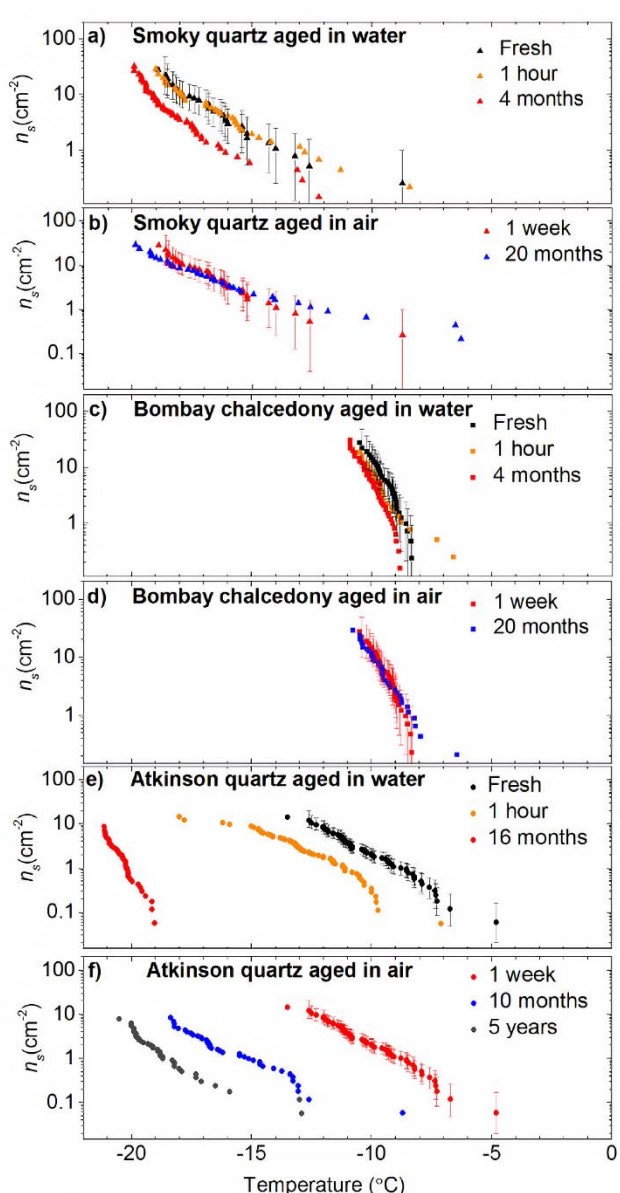


**Figure 4:** Plots showing the sensitivity of quartz activity, expressed as $n_s$, to time spent in water and air. Data are shown for **(a and b)** Smoky quartz, **(c and d)** Bombay chalcedony and **(e and f)** Atkinson quartz. Error bars for the first run of each time series are shown, but omitted for the other datasets for clarity.. The $n_s$ values for the fresh (~10 min) and one hour suspensions were taken from Fig. 2.




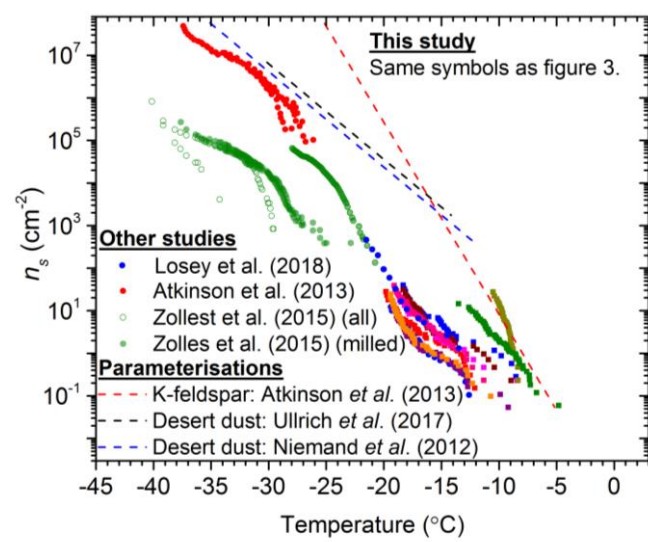


**Figure 5:** Plot of $n_s$ versus temperature for the available literature data for quartz compared to the data collected in this study. The symbols for this study's data are displayed the same as in Fig. 2 and only the first runs (fresh samples) from this study are plotted. The data from Zolles et al. (2015) has been split into quartz samples which were milled for fresh surfaces and all the combined data (both milled and un-milled quartz).






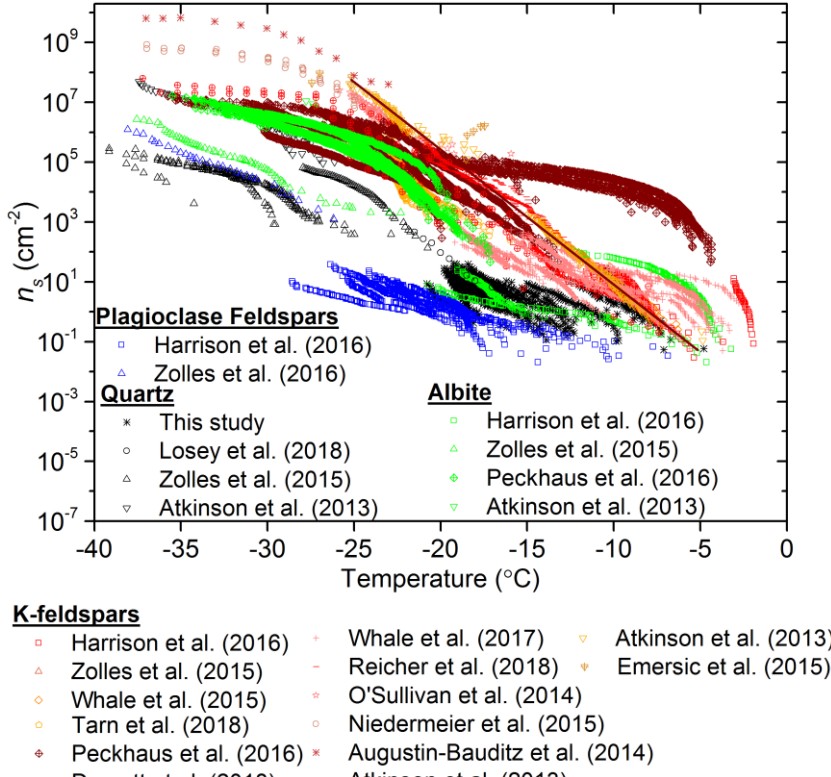

**Figure 6:** Plot of $n_s$ versus temperature for quartz and feldspar literature data, together with the quartz data from this study.


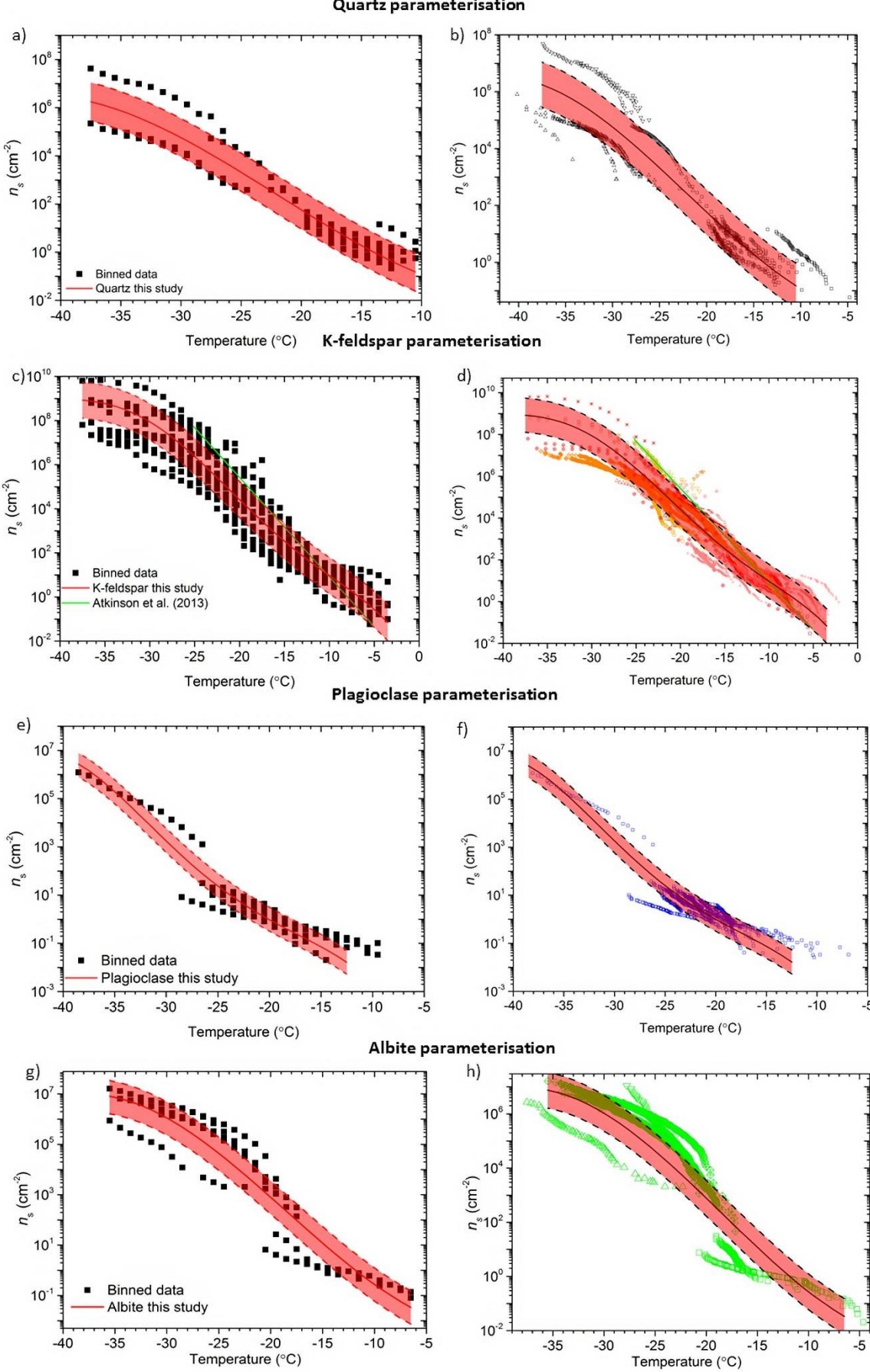

**Figure 7:** Parameterisations developed for various silicate minerals. **(a)** Temperature binned data for quartz which was used
to calculate the parameterisation with the equation $\log(n_s(T)) = -1.709 + (2.66\text{E-}4T^3) + (1.75\text{E-}2T^2) + (7\text{E-}2T)$, valid in the
range of -10.5 to -37.5 ˚C with a standard deviation of ± 0.8. **(b)** The newly developed parameterisation plotted over the raw
quartz data. **(c)** Temperature binned data for K-feldspar which was used to calculate the parameterisation with the equation
$\log(n_s(T)) = -3.25 + (-7.93\text{E-}1T^1) + (-6.91\text{E-}2T^2) + (-4.17\text{E-}3T^3) + (-1.05\text{E-}4T^4) + (-9.08\text{E-}7T^5)$, valid in the range of -3.5 to -
37.5 ˚C with a standard deviation of ± 0.8. **(d)** The newly developed parameterisation plotted over the raw K-feldspar data.
**(e)** Temperature binned data for plagioclase feldspars which was used to calculate the parameterisation with the equation
$\log(n_s(T)) = (-3.24\text{E-}5T^4) + (-3.17\text{E-}3T^3) + (-1.06\text{E-}1T^2) + (-1.71T)-12$, valid in the range of -12.5 to -38.5 ˚C with a standard
deviation of ± 0.5. **(f)** The newly developed parameterisation plotted over the raw plagioclase data. **(g)** Temperature binned
data for albite which was used to calculate the parameterisation with the equation $\log(n_s(T)) = (3.41\text{E-}4T^3) + (1.89\text{E-}2T^2) + (-$
$1.79\text{E-}2T)-2.29$, valid in the range of -6.5 to -35.5 ˚C with a standard deviation of ± 0.7. **(h)** The newly developed
parameterisation plotted over the raw albite data. The standard deviation is highlighted in the red shaded area for each
parameterisation and data considered to be unrepresentative of the bulk is excluded from the raw data.



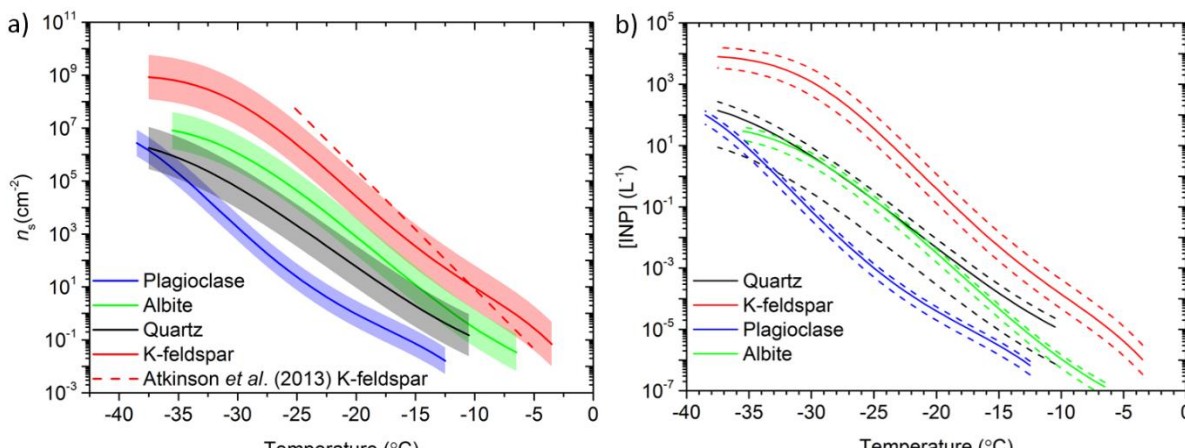

**Figure 8:** Comparison of the newly developed parameterisations. **(a)** $n_s$ versus temperature for the four newly created
parameterisations from this study and the K-feldspar parameterisation proposed by Atkinson et al. (2013). The standard
deviation of each parameterisation is shown by the shaded regions. **(b)** INP concentration per litre predictions using the
quartz, K-feldspar, albite and plagioclase parameterisations. The solid lines represent the predicted INP concentration
associated with the average mineral proportion and the dashed lines represent the upper and lower proportions based on the
variability of mineral proportions of atmospheric desert dust (see text for details). An aerosol surface area concentration of
50 µm$^2$ cm$^{-3}$ and an external mixing assumption were used.

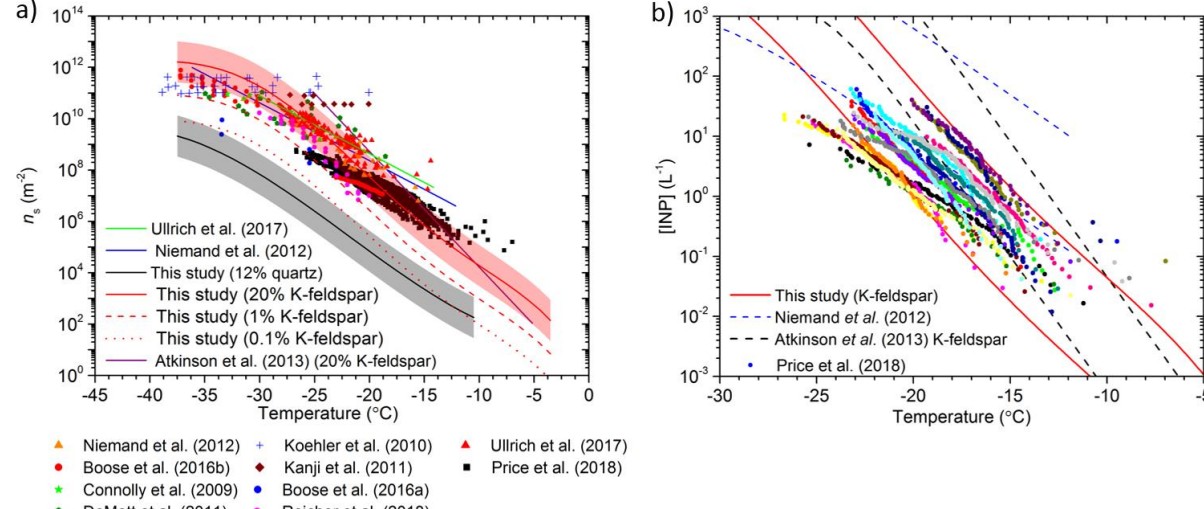

 **Figure 9:** Testing the newly developed K-feldspar and quartz parameterisations against literature data for desert dust. **a)**
Comparison of $n_s$ versus temperature for mineral dust from laboratory and field studies against the K-feldspar and quartz
parameterisations. The red lines are $n_s$ values where 0.1, 1 and 20 % of the aerosol surface area is assumed to be K-feldspar.
The standard deviation of the K-feldspar parameterisation from this study is represented as the shaded area around the 20 %
K-feldspar prediction: this is to show the natural variability in mineral activity. The prediction for 12 % quartz is shown
using a black line, with the natural mineral variability of freshly milled quartz highlighted by the shaded region. Literature
data and parameterisations have been plotted from (Niemand et al., 2012;Boose et al., 2016b;Boose et al., 2016a;Connolly et
al., 2009;DeMott et al., 2011;Koehler et al., 2010;Kanji et al., 2011;Reicher et al., 2018;Ullrich et al., 2017;Price et al.,
2018). **b)** Comparison of the INP concentrations predicted by several parameterisations with the INP concentrations
measured in the dusty eastern tropical Atlantic region by Price et al. (2018). The predictions were made assuming that 20%
of the dust was K-feldspar, consistent with Kandler et al. (2011). For this calculation we assumed that the dust is externally
mixed in terms of its mineralogy, although in this regime an internal versus external mixing state assumption makes very
little difference (see Atkinson et al. (2013)). The upper and lower bounds of the predicted INP concentrations are based on
the lowest and highest aerosol surface area concentrations corresponding to the INP data in Price et al. (2018). Note that the
measured INP concentrations from Price et al. (2018) have been corrected downwards by a factor of 2.5 based on the work
presented by Price et al. (2018) and Sanchez-Marroquin et al. (2019).