# Peer review of "The ice-nucleating ability of quartz immersed in water and its"

_Atmospheric Chemistry and Physics, 2019_

## Referee Comment (RC1) · Anonymous Referee #1 · 22 May 2019

**Review of "The ice-nucleating ability of quartz immersed in water and its atmospheric importance compared to K-feldspar" by Harrison et al.**

**General comment**

This study investigated the ice-nucleating abilities of 10 different $\alpha$-quartz samples and compare the obtained results with feldspar literature data. The authors found a large variability in the ice nucleation behavior of the 10 samples with some being very efficient (e.g., Bombay chalcedony and Atkinson quartz). It was also found that the ice-nucleating abilities of some of the studied quartz samples were affected by ageing and milling. With the current and literature data the authors produced a new set of parametrizations for quartz and different feldspar. The authors found that the new K-feldspar parameterisation provides a good representation of the ice-nucleating activity of dust from field and laboratory studies. Finally, their analysis show that quartz is of second order importance for desert dust's ice-nucleating ability.

This is a well written and sound manuscript with interesting results for the ice nucleation community. The experiments were well designed and were properly executed. The paper nicely fits with the ACP scope and it can be accepted for its publication after the following points are properly addressed.

**Major Comment**

Although I really like the manuscript, I am having a hard time to find the atmospheric relevance of the obtained results. Given that milling is a process that does not takes place in nature the way it was conducted in the laboratory, I am not fully convinced that the obtained parametrisations can really be used in climate models as the representation of true atmospheric mineral dust particles. Similarly, I also found that the very long ageing times (i.e., > several months) are not atmospherically relevant.

**Minor comments**

1. The title states: "The ice-nucleating ability of quartz immersed in water and **its atmospheric importance** compared to K-feldspar". Can the laboratory experiments be assume to be relevant to the atmosphere? Does milling takes place in the atmosphere and soil the way it was done in the laboratory?
2. The abstract is very descriptive without quantitative data. I suggest to add the most important quantitative results here.
3. The authors indicate that the base line was obtained from Umo et al. (2015). Does it mean that you did not run these type of experiments prior to the heterogeneous ice nucleation ones? How confident are the authors that this did not change since 2015? I suggest to add your own data and to remove the Umo et al. (2015).
4. Section 4.2: Is milling atmospherically relevant at all? Can this happen in nature to this extent? Please motivate this deeply.

5. Conclusions: Although the authors provide potential explanations for their observations, several sentences/conclusions seems to be speculative.
6. References: Add the Doi to all references. The journal names must be abbreviated. I find exaggerated to have 20 citations from the same research group.

**Technical comments**

Line 25-26: for ice nucleation.
Line 27: "less active". By how much?
Line 29: "more active". By how much?
Line 51: Add a reference after "incomplete".
Line 54: "Observations of aerosol at the centre of ice crystals have shown that mineral dust". Is it really at the centre?
Lines 54-56: Either state that this sentence focuses on MPC only or add other studies such as Cziczo et al. (2013).
Lines 61-62: I suggest to add the review of Hoose and Mohler (2012) given that this list is too short to reflect the huge amount of work done with clays.
Line 117: Add a reference after "dust aerosol".
Lines 185-186: "and a study of nitric acid hydrate nucleation on meteoric material (James et al., 2018)." It does not fit here.
Line 188: "was vigorously shaken". Manually?
Lines 119-121: "An almost identical method was described by Harrison et al. (2016), which was similar to the work of Wright and Petters (2013)." Unnecessary self-citation.
Line 243: Delete "approximately 1 h later" It is redundant.
Lines 245-246: "There is also consistency for some quartz samples between run to run from this study". This was already mentioned seven lines above: "In the cases of Bombay chalcedony, Brazil amethyst and Smokey quartz, the first and second runs where identical within the uncertainties".
Line 295: "sealed glass vial for". Under dark conditions?
Line 309: "a dark cupboard in sealed glass vials". Why under dark conditions? What atmospheric process does it represent?
Line 316: "by about 3 °C". Do the authors consider this a significant change?
Line 321: "by 2 °C". Do the authors consider this a significant change?
Line 321: "after 16 months in water". Is this atmospherically relevant?
Line 326: "for 20 months". Is this atmospherically relevant?
Line 329: "for ~5 years in a glass vial". Is this atmospherically relevant?
Line 544-545: "quartz, but the parameterisation we present here probably represents an upper limit to its activity." I am wondering if the ice-nucleating abilities of the quartz samples are unintentionally overestimated by milling.

---

## Referee Comment (RC2) · Anonymous Referee #2 · 23 May 2019

**General Comment**

The manuscript I was asked to evaluate is dedicated to the experimental study of immersion freezing behavior of several α-quartz samples in pure water droplets. The authors highlight the variability of the ice nucleation activity over different freshly milled quartz samples and investigate the short term as well as long term aging effects due to exposure to air and water. They further propose active site density parameterizations for several minerals and discuss the dominance of K-feldspar in the ice nucleation particle population in desert dusts. While I support the publication, I do have few remarks that the authors should address while preparing the final version of the manuscript.

**Major comments:**

Line 387 It should rather be "….parameterisation is representative of freshly milled quartz dust". Airborne dust is eroded and gets exposed to air and even water. The authors should add a discussion about how representative is this for airborne dust. Proposing such parameterization - especially for 'freshly milled' quartz – is a bit of a stretch for now, given that we barely understand the surface due to limited work done and the high variability in its ice nucleation efficiency reported so far (including this work).

It should be clearly stated in the abstract that the study is on "freshly milled quartz" particles (also throughout the text) and should include a comment on the atmospheric relevance of such fresh surfaces when drawing comparisons with atmospheric dusts.

There are instances in Discussion section about cautiously using plagioclase and albite parameterization, yet this feature does not translate in the figures. In general, the devised parameterizations are a result of over-simplification and multiple assumptions which, though well-fitting, might not give a comprehensive view.

Given that members of K-feldspars show huge variations in their ice nucleation ability, with only microcline as a standout for most part (Harrison et al., 2016; Kaufmann et al., 2016; Welti et al., 2019), would the authors say that the K-feldspar parameterization proposed here is actually more representative for just microcline? In addition, following the information given in line 419-420, surprisingly recent data on natural dust mineralogy (Boose et al., 2016; Kaufmann et al., 2016) was not considered which may have painted a different picture.

Figure 2 & Methodology: It is unclear in the methodology whether the suspensions were tested just once or multiple times and why no uncertainties are shown in Fraction Frozen curves.

Figure 3: There are no error bars for freezing data of quartz samples undergone long-term aging in either water or exposed to air. Are these single suspension measurements? This should be made clear in the methodology section.

**Minor comments**

Line 129 Stone sample milling: average particle size range information of the samples could be helpful. I assume from the methodology that the stones were "hand-milled". Given that quartz is such a hard mineral, one might still end up with powders consisting of large particles which contribute more to the mass fraction when preparing suspensions. I would highly encourage the authors to consider adding a discussion on size of mineral particles typically found at/near source regions versus the size range of particles probed in the experiments and its atmospheric relevance.

Lines 139-141 Glass vials as suspension storage containers: Glass is a source of Si and other ionic contamination. If the suspension reaches supersaturation in Si-concentration with respect to quartz

surface, it is likely that the quartz surface will start to grow over longer time scales and affect the ice nucleation ability of quartz (Kumar et al., 2019a). This process is governed by several factors eg. Si-concentration, particle surface exposed, growth rate, etc. (Baughman, 1991), which does not seem to lead to similar deteriorating IN efficiency in 3 quartz samples tested here in this study. It would be good if the authors could comment on this or add a discussion.

Line 248-286 Could it be that the indifferent IN ability in cases of Bombay Chalcedony and Smoky quartz during aging in water compared to Atkinson quartz be due to lower particle surface area exposed (low BET hence slower aging effect) for former 2 samples? Would have been interesting to see the long-term aging of the other chalcedony (Grape) which has similar BET value as Atkinson quartz.

Line 293 what exactly is meant by "dangling OH groups"? Highly hydroxylated quartz surfaces are dominated by vicinal and germinal silanols (Muster et al. (2001) and references therein) which then tend to create network of H-bonds with each other (Musso et al., 2011; 2012), therefore, not really free and "dangling" per se.

**Technical comments:**

Since the manuscript exclusively talks about ice nucleating ability of various minerals, I would suggest the term "material/materials" be replaced by "mineral/minerals" throughout the manuscript (eg. line 19, 171, 207 etc.)

Line 24-28 This is a long sentence. This can be split into 2 to better convey the meaning, eg. 2nd sentence could be "the ice nucleation particle population in desert dust aerosol is dominated by K-feldspars rather than quartz (or other minerals)."

Line 41 "type of aerosol particles" instead of "aerosol types"

Line 43 "Field observations of ice crystal residuals" instead of "Observations of aerosol at the centre of ice crystals"

Line 44 Cite the accepted ACP version of Eriksen Hammer et al. (2018)

Line 46-49 May consider splitting this into "Atmospheric mineral dusts are composed of several components. Clay is the major component of airborne mineral dust and is sufficiently small that its atmospheric lifetime is relatively long. Hence recent ice nucleation studies have focused on the clay group of minerals."

Should be 'INPs' in place of 'ice nucleants' (line 51) and 'ice nucleating material' (line 71)

Line 67 'repeated' instead of 'repeat'

Line 70 remove 'quite'

Line 74 "over time when suspended in water or exposed to air" would be better in place of 'to time spent in water or air'

Line 104 $TiO_2$ is mentioned twice. Remove one of them

Line 139 this line gives the impression that the samples undergoing BET tests were used to make suspensions, which I believe is not the case. Consider removing "After BET analysis"

Line 176-177 delete 'specific' from line 176 and add it before "surface area" in line 177

Line 293-295 give references after 'configuration'

Line 332 "It may be these microtextural differences that lead to the observed variability in ice-nucleating ability"

Line 377-379 Consider re-phrasing the lines to "However, we constrained the polynomial fits because the unconstrained fits poorly represented the data at the warmest and coldest ends." for easy understanding.

Line 415 "its exceptionally high ice nucleation ability" in place of 'it exhibiting exceptional behaviour'

Line 447 can also add Kumar et al. (2018); Kumar et al. (2019b) in the references in regards to aging of K-feldspar microcline

Line 471-472 Fig 8b shows parameterisation for desert dust by Niemand et al. (2012) in blue dashed lines and the K-feldspar parameterisation proposed in this study by solid red lines

Lines 506-510 Weathering in solutes has already been addressed by Whale et al. (2018) and 3 Parts series from Kumar et al. and should be addressed as part of this paper rather than referring to future work

Line 642, 666 and 673 - cite the accepted ACP version of these papers: Kumar et al. (2019a), Peckhaus et al. (2016), Pinti et al. (2012)

Figure 2 add suspension concentration in caption

Figure 5 Niedermeier et al. (2015) - correct author name in figure legend. Also, the curves are difficult to read, especially with orange-red tones, maybe use other dark colors. Size of symbols in legend needs to be increased for better readability

Figure 8 Check references to Boose et al 2016 (for 'a' and/or 'b') for data used in the figure. Symbol size in legend should be increased. In line 826 "….the natural mineral variability of freshly milled quartz highlighted…."

Line 795-796 equation formatting needs improvement

**References**

Baughman, R. J.: Quartz crystal growth, Journal of Crystal Growth, 112, 753-757, doi:10.1016/0022-0248(91)90132-O, 1991.

Boose, Y., Welti, A., Atkinson, J., Ramelli, F., Danielczok, A., Bingemer, H. G., Plötze, M., Sierau, B., Kanji, Z. A., and Lohmann, U.: Heterogeneous ice nucleation on dust particles sourced from nine deserts worldwide – Part 1: Immersion freezing, Atmos. Chem. Phys., 16, 15075-15095, doi:10.5194/acp-16-15075-2016, 2016.

Eriksen Hammer, S., Mertes, S., Schneider, J., Ebert, M., Kandler, K., and Weinbruch, S.: Composition of ice particle residuals in mixed-phase clouds at jungfraujoch (switzerland): Enrichment and depletion of particle groups relative to total aerosol, Atmos. Chem. Phys., 18, 13987-14003, doi:10.5194/acp-18-13987-2018, 2018.

Harrison, A. D., Whale, T. F., Carpenter, M. A., Holden, M. A., Neve, L., O'Sullivan, D., Vergara Temprado, J., and Murray, B. J.: Not all feldspars are equal: A survey of ice nucleating properties across the feldspar group of minerals, Atmos. Chem. Phys., 16, 10927-10940, doi:10.5194/acp-16-10927-2016, 2016.

Kaufmann, L., Marcolli, C., Hofer, J., Pinti, V., Hoyle, C. R., and Peter, T.: Ice nucleation efficiency of natural dust samples in the immersion mode, Atmos. Chem. Phys., 16, 11177-11206, doi:10.5194/acp-16-11177-2016, 2016.

Kumar, A., Marcolli, C., Luo, B., and Peter, T.: Ice nucleation activity of silicates and aluminosilicates in pure water and aqueous solutions – Part 1: The K-feldspar microcline, Atmos. Chem. Phys., 18, 7057-7079, doi:10.5194/acp-18-7057-2018, 2018.

Kumar, A., Marcolli, C., and Peter, T.: Ice nucleation activity of silicates and aluminosilicates in pure water and aqueous solutions – part 2: Quartz and amorphous silica, Atmos. Chem. Phys., 19, 6035-6058, doi:10.5194/acp-19-6035-2019, 2019a.

Kumar, A., Marcolli, C., and Peter, T.: Ice nucleation activity of silicates and aluminosilicates in pure water and aqueous solutions – part 3: Aluminosilicates, Atmos. Chem. Phys., 19, 6059-6084, doi:10.5194/acp-19-6059-2019, 2019b.

Musso, F., Ugliengo, P., and Sodupe, M.: Do H-Bond features of silica surfaces affect the H2O and NH3 adsorption? Insights from periodic B3LYP calculations, J. Phys. Chem. A, 115, 11221-11228, doi:10.1021/jp203988j, 2011.

Musso, F., Mignon, P., Ugliengo, P., and Sodupe, M.: Cooperative effects at water–crystalline silica interfaces strengthen surface silanol hydrogen bonding. An ab initio molecular dynamics study, Phys. Chem. Chem. Phys., 14, 10507-10514, doi:10.1039/C2CP40756F, 2012.

Muster, T. H., Prestidge, C. A., and Hayes, R. A.: Water adsorption kinetics and contact angles of silica particles, Colloids and Surfaces A: Physicochemical and Engineering Aspects, 176, 253-266, doi:https://doi.org/10.1016/S0927-7757(00)00600-2, 2001.

Peckhaus, A., Kiselev, A., Hiron, T., Ebert, M., and Leisner, T.: A comparative study of K-rich and Na/Ca-rich feldspar ice-nucleating particles in a nanoliter droplet freezing assay, Atmos. Chem. Phys., 16, 11477-11496, doi:10.5194/acp-16-11477-2016, 2016.

Pinti, V., Marcolli, C., Zobrist, B., Hoyle, C. R., and Peter, T.: Ice nucleation efficiency of clay minerals in the immersion mode, Atmos. Chem. Phys., 12, 5859-5878, doi:10.5194/acp-12-5859-2012, 2012.

Welti, A., Lohmann, U., and Kanji, Z. A.: Ice nucleation properties of K-feldspar polymorphs and plagioclase feldspars, Atmos. Chem. Phys. Discuss., 2019, 1-25, doi:10.5194/acp-2018-1271, 2019.

Whale, T. F., Holden, M. A., Wilson, Theodore W., O'Sullivan, D., and Murray, B. J.: The enhancement and suppression of immersion mode heterogeneous ice-nucleation by solutes, Chemical Science, 9, 4142-4151, doi:10.1039/C7SC05421A, 2018.

---

## Author Comment (AC1) · 17 Jul 2019

**Response to referee comments**

We would like to thank both referees for their insightful comments. Below we have addressed the referee comments (in red) with our responses in black and changes made to the manuscript in blue.

**Response to referee #1**

**General comment**

This study investigated the ice-nucleating abilities of 10 different $\alpha$-quartz samples and compare the obtained results with feldspar literature data. The authors found a large variability in the ice nucleation behaviour of the 10 samples with some being very efficient (e.g., Bombay chalcedony and Atkinson quartz). It was also found that the ice-nucleating abilities of some of the studied quartz samples were affected by ageing and milling. With the current and literature data the authors produced a new set of parametrizations for quartz and different feldspar. The authors found that the new K-feldspar parameterisation provides a good representation of the ice-nucleating activity of dust from field and laboratory studies. Finally, their analysis show that quartz is of second order importance for desert dust's ice-nucleating ability.

This is a well written and sound manuscript with interesting results for the ice nucleation community. The experiments were well designed and were properly executed. The paper nicely fits with the ACP scope and it can be accepted for its publication after the following points are properly addressed.

**Major Comment**

Although I really like the manuscript, I am having a hard time to find the atmospheric relevance of the obtained results. Given that milling is a process that does not takes place in nature the way it was conducted in the laboratory, I am not fully convinced that the obtained parametrisations can really be used in climate models as the representation of true atmospheric mineral dust particles.

We note that this comment on the relevance of milling is similar to referee 2, who asks us to motivate the use of milled samples.

In nature, the path from rock through to a fine dust powder which can be aerosolised is complex and the referee is correct to question the relevance of simply taking a rock sample and milling it. The assumption in this study (and others from groups around the world) is the ice-nucleating ability of the mineral samples is inherent to the material and that milling is representative of the mechanical processes which produce fine airborne dust in nature. We argue this on the basis that the processes which produce atmospheric dusts are mechanical and energetic. Rocks are broken down into soil particles through mechanical and chemical weathering processes and also dust is aerosolised through the saltation process which involves impactions of particles which are aggressive enough to round grains and cause fragmentation of

aggregates. With mechanical processes occurring in nature we think that milling of macroscopic rock samples to fine grain sizes is relevant. This approach has the advantage that we can obtain relatively pure minerals in the form of rocks or large crystals; whereas soils are mixtures of many minerals.

There are of course processes which will occur in nature that are likely to change the activity of natural dusts since they are exposed to air and water for many years prior to aerosolisation. In general, ageing processes are thought to reduce the ice-nucleating activity of freshly ground powders, hence we make the assumption that milled samples are likely to exhibit a maximum activity and this may decrease with ageing. We have found that water and air exposure does not substantially decrease the activity of K-feldspar (Harrison et al., 2016;Whale et al., 2017) (although acid may be another story (Kumar et al. (2018)). However, quartz shows a strong sensitivity to time spent in water and so we stress in the paper that the quartz parameterisation is likely an upper limit representation of the ice-nucleating ability of quartz. On the other hand, the saltation aerosolisation process is mechanically vigorous and may expose fresh surfaces, hence taking this upper limit is reasonable. When taking this upper limit for quartz it is still seen to be only of second order importance in terms of its contribution to the atmospheric INP population relative to K-feldspars. Hence we think our conclusion regarding the atmospheric importance of quartz relative to K-feldspar is justified.

We then went on to compare the predicted INP concentrations based on our K-feldspar and quartz parameterisations to atmospheric measurements in this paper and find that we get a reasonable prediction of the INP concentration based the on K-feldspar parameterisation. This is consistent with the milled K-feldspar samples from the literature being of relevance to the atmosphere.

The other point we would like to make is that it is not clear how else these studies could be done. One approach might be to take desert dust samples and examine their ice-nucleating ability (as we and other groups have done). While this approach has value (and should be used) it has a number of caveats associated with it too:

1. Without knowledge of which components of desert dust are causing it to nucleate ice, predictive capacity of the ice-nucleating activity of desert soils from different regions with different compositions is limited.
2. Without knowledge of which components nucleate ice we cannot predict how desert dust's activity will evolve as the largest particles are removed on transport (mineralogy is size dependent with K-feldspars and quartz being preferentially in the larger size range).
3. Sampling and re-suspension of desert dusts may influence its ice-nucleating ability. Mechanical aerosolisation processes, such as rotating brushes, used to resuspend dust may well change the surface properties of desert dust particles in a manner different to the way dust is altered in the saltation process.

Hence, we think that in order to gain a predictive and accurate quantitative description of ice nucleation by atmospheric mineral dusts it is necessary to do experiments with both desert dusts and also the individual minerals in a relatively pure state as well as conduct experiments examining the sensitivity of these dusts to ageing processes.

To address this issue in the paper we have added a new paragraph in the introduction and a new schematic diagram, as well as a brief statement in the methods section stating: 'These samples were reground to ensure all samples initially had freshly exposed surfaces for ice nucleation experiments. The milling process was used to break down mineral crystals/powders to a sufficient size so that they may be suspended in water. We argue that these freshly milled samples are relevant in that they represent the fresh surfaces which are likely produced by mechanical processes in nature as rocks are broken down and particles aerosolised through the saltation process (see Figure 1 and discussion in the introduction). We suggest that these freshly ground samples of quartz represent an upper limit to the ice-nucleating ability of quartz in atmospheric mineral dust since ageing processes may reduce this activity.'

Similarly, I also found that the very long ageing times (i.e., > several months) are not atmospherically relevant.

We think that these times are relevant for dust in the atmosphere, because the dust on the ground is exposed to water and air for very long periods of time. Hence, exposure to air and water for many months was a pragmatic semi-quantitative way of assessing the sensitivity of these active sites to ageing processes. We think the new figure and the new paragraph in the introduction help to clarify this.

**Minor comments**

1. The title states: "The ice-nucleating ability of quartz immersed in water and its atmospheric importance compared to K-feldspar". Can the laboratory experiments be assumed to be relevant to the atmosphere? Does milling takes place in the atmosphere and soil the way it was done in the laboratory?

We have addressed this in the above response to the major comment.

2. The abstract is very descriptive without quantitative data. I suggest to add the most important quantitative results here.

The abstract has been amended with some pertinent numbers to illustrate the key points.

3. The authors indicate that the base line was obtained from Umo et al. (2015). Does it mean that you did not run these type of experiments prior to the heterogeneous ice

nucleation ones? How confident are the authors that this did not change since 2015? I suggest to add your own data and to remove the Umo et al. (2015).

The Umo et al. (2015) baseline fit was developed from a large compilation of baseline measurements to incorporate the variability within the µL-NIPI technique. This fit provides a convenient yet robust expression of the variability within this experimental procedure. Baseline experiments were conducted prior to the start of each experimental day with the fraction frozen curves typically being represented by the lower range of freezing values presented by Umo et al. (2015). With this being said we have used the Umo et al. (2015) parameterisation to show the variability of the experiment with greater statistics and to show a conservative estimate of the background (in general our baseline has improved since 2015). As can be seen from figure 2 it is observable that all experiments provided freezing events that are well above the baseline even with the more conservative representation by Umo et al. (2015). The text in the methodology has been amended to read:

"Prior to the start of each experimental day purified water droplets were used to determine the background freezing signal. Umo et al. (2015) compiled a large collection of background freezing results to create a fit which represents the variability of the background in the µL-NIPI instrument. The background signal measured in this study was in line with the lower bound set by Umo et al. (2015)."

4. Section 4.2: Is milling atmospherically relevant at all? Can this happen in nature to this extent? Please motivate this deeply.

Please see response to major comments above.

5. Conclusions: Although the authors provide potential explanations for their observations, several sentences/conclusions seems to be speculative.

We have improved the conclusions section by adding additional references and making it clearer what new work needs to be done in the future.

We have amended the paragraph on ageing to read:

"Related to this, we also note that solutes can alter the ice-nucleating ability of mineral samples (Whale et al., 2018;Kumar et al., 2018;Kumar et al., 2019a, b). Sensitivity to these ageing processes and solutes could be very important in determining the dominant INP types globally (Boose et al. 2019). Hence, we suggest further studies aim to build a better understanding of the relationship between the experimental observations and field collected samples to determine the role of ageing in the atmosphere."

We have added:

"Sparse data sets available for the albite and plagioclase mineral groups lead to lower confidence when creating parameterisations for these mineral groups. It is suggested that future studies expand on the current datasets of the ice-nucleating behaviour of minerals to improve these parameterisations."

And clarified that the quartz parameterisation should be thought of as an upper limit:

"Also note that the parameterisation for quartz is for freshly milled quartz and the ageing results presented here and elsewhere (Zolles et al., 2015;Kumar et al., 2019a) suggest that the active sites on quartz are removed on exposure to air and water. Therefore the parameterisation for milled quartz should be regarded as an upper limit. Even with this upper limit, quartz is of secondary importance relative to K-feldspars which appear to be less sensitive to ageing processes."

6. References: Add the Doi to all references. The journal names must be abbreviated. I find exaggerated to have 20 citations from the same research group.

We have amended the reference list as suggested.

We have removed James et al. But, we think that the other references are all necessary and justified. Our research group has contributed significantly to this area of research, hence the number of citations.

**Technical comments**

Line 25-26: for ice nucleation. Now corrected.

Line 27: "less active". By how much? "…generally less active than K-feldspars by roughly 7 °C"

Line 29: "more active". By how much? "…the quartz samples are generally more active by roughly 5 °C"

Line 51: Add a reference after "incomplete". Amended to "However, our understanding of which type of aerosol particles serve as effective INPs is incomplete (Vergara-Temprado et al., 2017;Kanji et al., 2017)."

Line 54: "Observations of aerosol at the centre of ice crystals have shown that mineral dust". Is it really at the centre? "Observations of aerosol within ice crystals"

Lines 54-56: Either state that this sentence focuses on MPC only or add other studies such as Cziczo et al. (2013). "…act as INPs within mixed phase clouds"

Lines 61-62: I suggest to add the review of Hoose and Mohler (2012) given that this list is too short to reflect the huge amount of work done with clays. Now added.

Line 117: Add a reference after "dust aerosol". Added "(Deer et al., 1992)".

Lines 185-186: "and a study of nitric acid hydrate nucleation on meteoric material (James et al., 2018)." It does not fit here. The sentence has been amended to read "This technique has been used in several previous ice nucleation studies e.g. (Atkinson et al., 2013;O'Sullivan et al., 2014;Harrison et al., 2016) ."

Line 188: "was vigorously shaken". Manually? Thank you, this has been amended to "…the suspension was vigorously shaken manually"

Lines 119-121: "An almost identical method was described by Harrison et al. (2016), which was similar to the work of Wright and Petters (2013)." Unnecessary self-citation. Changed to: "This methodology was based on the work of Wright and Petters (2013)."

Line 243: Delete "approximately 1 h later" It is redundant. This has been deleted.

Lines 245-246: "There is also consistency for some quartz samples between run to run from this study". This was already mentioned seven lines above: "In the cases of Bombay chalcedony, Brazil amethyst and Smokey quartz, the first and second runs where identical within the uncertainties". The sentence has been removed

Line 295: "sealed glass vial for". Under dark conditions? "…sealed glass vial under dark conditions for…"

Line 309: "a dark cupboard in sealed glass vials". Why under dark conditions? What atmospheric process does it represent? The aim of this time series investigation was not to simulate the atmosphere – see comments above. We have simply described what we did.

Line 316: "by about 3 °C". Do the authors consider this a significant change? Text has been amended to read "its activity decreased by about 3 °C after four months in water which is well outside the uncertainties of the experiment."

Line 321: "by 2 °C". Do the authors consider this a significant change? This is addressed in the previous sentence in the text "In contrast, the activity of Atkinson quartz decreased dramatically"

Line 321: "after 16 months in water". Is this atmospherically relevant?

We have addressed this above.

Line 326: "for 20 months". Is this atmospherically relevant?

We have addressed this above.

Line 329: "for ~5 years in a glass vial". Is this atmospherically relevant?

We have addressed this above. It is certainly a relevant time period for dust sitting on the surface. In this case, 5 years was simply the time the sample had been in the lab which we opportunistically made use of.

Line 544-545: "quartz, but the parameterisation we present here probably represents an upper limit to its activity." I am wondering if the ice-nucleating abilities of the quartz samples are unintentionally overestimated by milling. This issue is very much related to the 1st major comment on milling, which we have now addressed much more thoroughly. The short answer, is yes, the quartz parameterisation based on freshly milled material may be an overestimate. This is why we refer to it as an

upper limit. We have made this clearer throughout the paper, including in the conclusions.

**Response to referee #2**

**General Comment**

The manuscript I was asked to evaluate is dedicated to the experimental study of immersion freezing behaviour of several α-quartz samples in pure water droplets. The authors highlight the variability of the ice nucleation activity over different freshly milled quartz samples and investigate the short term as well as long term aging effects due to exposure to air and water. They further propose active site density parameterizations for several minerals and discuss the dominance of K-feldspar in the ice nucleation particle population in desert dusts. While I support the publication, I do have few remarks that the authors should address while preparing the final version of the manuscript.

**Major Comments**

Line 387 It should rather be "….parameterisation is representative of freshly milled quartz dust". Airborne dust is eroded and gets exposed to air and even water. The authors should add a discussion about how representative is this for airborne dust. Proposing such parameterization - especially for 'freshly milled' quartz – is a bit of a stretch for now, given that we barely understand the surface due to limited work done and the high variability in its ice nucleation efficiency reported so far (including this work).

This issue has been also been raised by referee 1 and we have now included a new paragraph in the introduction and methods to justify our use of milled samples and how these samples relate to the complex processes dust experiences in nature. Please see the response to the major comment by reviewer 1.

We have changed Line 387 as recommended.

It should be clearly stated in the abstract that the study is on "freshly milled quartz" particles (also throughout the text) and should include a comment on the atmospheric relevance of such fresh surfaces when drawing comparisons with atmospheric dusts.

Done

There are instances in Discussion section about cautiously using plagioclase and albite parameterization, yet this feature does not translate in the figures. In general, the devised parameterizations are a result of over-simplification and multiple assumptions which, though wellfitting, might not give a comprehensive view.

The parameterisations are created based on the current available data. Although crude, these parameterisations provide evidence that K-feldspar is the dominant mineral within mineral dusts which is supported by the good agreement between

field observations and model predictions using lab based K-feldspar predictions. We have now added to the conclusions section to outline the importance of advancing our current datasets for mineral ice-nucleating abilities:

"Sparse data sets available for the albite and plagioclase mineral groups lead to lower confidence when creating parameterisations for these mineral groups. It is suggested that future studies expand on the current datasets of the ice-nucleating behaviour of minerals to improve these parameterisations"

Given that members of K-feldspars show huge variations in their ice nucleation ability, with only microcline as a standout for most part (Harrison et al., 2016; Kaufmann et al., 2016; Welti et al., 2019), would the authors say that the K-feldspar parameterization proposed here is actually more representative for just microcline?

We think it is very important that we provide a parameterisation for K-feldspar in general. We base this on the finding that microtexture is important, not the subtle differences in crystal structure which give rise to the three K-feldspars (microcline, sanidine and orthoclase). Microcline has an ordered Al position, whereas sanidine is disordered with orthoclase having some intermediate disorder. The distinction between these polymorphs is therefore somewhat arbitrary with X-ray diffraction and Raman techniques since there is a continuum in ordering of Al.

The work by Harrison et al. (2016), Whale et al. (2017) and Welti et al. (2019) show that orthoclase nucleates ice in the same regime as microcline (in most instances). How the nucleating ability of sanidine compares to microcline and orthoclase is more complicated. There are examples of sanidine that are less active than microcline/orthoclase as an ice-nucleator (whale et al. 2017, Welti et al. 2019) and those that nucleate ice in the same regime (Harrison et al. 2016). This is interesting and maybe a function of the formation processes. Sanidine is a high temperature K-feldspar and so can cool much faster than other K-feldspar polymorphs and so could exhibit phase separation to a lesser extent.

With this said, sanidine is rarely observable in atmospheric mineral dusts (Boose et al. 2016) and so the parametrisation is a fair representation of the K-feldspars for the atmosphere and is supported by the agreement with field observations of atmospheric dusts (Atkinson et al., 2013; Price et al., 2018; O'Sullivan et al., 2018).

The 4[th] paragraph in section 5.2 has been modified to clarify these points:

"The data included in these plots includes all three polymorphs of K-feldspar (microcline, orthoclase and sanidine), although most of the data is for microcline. The strongly hyperactive TUD #3, examined by Harrison et al. (2016) and Peckhaus et al. (2016), was excluded as it exhibited extremely high activity and appears to be an exceptional case which is generally unrepresentative of the K-feldspar group of minerals. With this in mind we have developed a parameterisation which represents K-feldspars that possess exsolution microtexture."

In addition, following the information given in line 419-420, surprisingly recent data on natural dust mineralogy (Boose et al., 2016; Kaufmann et al., 2016) was not considered which may have painted a different picture.

The values quoted in Boose et al. (2016) for transported dust mineralogy are consistent with those in Atkinson et al. (2013). Kaufmann et al. (2016) looked at surface scooped samples only, whereas we have taken values for airborne dust subject to transport. A reference to Boose et al. (2016) has been added in section 5.2.

Figure 2 & Methodology: It is unclear in the methodology whether the suspensions were tested just once or multiple times and why no uncertainties are shown in Fraction Frozen curves. This has been made clearer in section 3.2 "A second run for each sample suspension, with a fresh array of droplets, was performed immediately after the first experiment with approximately 1 hour between the two runs." The temperature uncertainty for the fraction frozen plots is quoted in the text as 0.4 K, but not shown in the figures for clarity.

Figure 3: There are no error bars for freezing data of quartz samples undergone long-term aging in either water or exposed to air. Are these single suspension measurements? This should be made clear in the methodology section.

The errors are only shown for one experiment in order to present the data more clearly. The errors do not significantly change from run to run as there is the same surface area of nucleator per droplet (as we use the same suspension) and the number of droplets is similar in each run. This has been made clearer in figure caption 3. "Error bars for the first run of each time series are shown, but omitted for the other datasets for clarity."

**Minor Comments**

Line 129 Stone sample milling: average particle size range information of the samples could be helpful. I assume from the methodology that the stones were "hand-milled". Given that quartz is such a hard mineral, one might still end up with powders consisting of large particles which contribute more to the mass fraction when preparing suspensions. I would highly encourage the authors to consider adding a discussion on size of mineral particles typically found at/near source regions versus the size range of particles probed in the experiments and its atmospheric relevance.

We make the assumption that ice nucleation scales with surface area, and report the ice active sites per unit surface area. We also make this assumption when considering the atmosphere. Whether the density of sites varies with particle size is an interesting question, but beyond the scope of this project. The BET surface areas are all within a factor of ~5, hence the mean particle size of the different samples is similar. Based on the BET surface area and a density of 2.65 g cm$^{-3}$ we expect a spherical equivalent average size of ~ 0.5 µm to 2.5 µm. Table 1 has been amended

to show these sizes and the new figure (figure 1) includes information on grain sizes in nature.

Lines 139-141 Glass vials as suspension storage containers: Glass is a source of Si and other ionic contamination. If the suspension reaches supersaturation in Si-concentration with respect to quartz surface, it is likely that the quartz surface will start to grow over longer time scales and affect the ice nucleation ability of quartz (Kumar et al., 2019a). This process is governed by several factors eg. Si concentration, particle surface exposed, growth rate, etc. (Baughman, 1991), which does not seem to lead to similar deteriorating IN efficiency in quartz samples tested here in this study. It would be good if the authors could comment on this or add a discussion.

We discuss Kumar's results in the second paragraph of section 4.3.

Line 248-286 Could it be that the indifferent IN ability in cases of Bombay Chalcedony and Smoky quartz during aging in water compared to Atkinson quartz be due to lower particle surface area exposed (low BET hence slower aging effect) for former 2 samples? Would have been interesting to see the long-term aging of the other chalcedony (Grape) which has similar BET value as Atkinson quartz.

This is an interesting concept and one we cannot definitively answer given our data. Grape chalcedony did show one of the largest sensitivities in the initial experiments (an hour left in water until the repeat run of the suspension). However, we do see that Bombay chalcedony does show less sensitivity to time left in water opposed to Smoky quartz (which has a similar surface area). We also see in the initial experiments that some quartz samples that had similar or lower surface areas than Bombay Chalcedony showed sensitivity (to time left in water) on the hourly time scale. This would indicate that the amount of surface area exposed cannot entirely explain the sensitivity of the samples and likely the nature of the sites also dictates the effect of ageing.

Line 293 what exactly is meant by "dangling OH groups"? Highly hydroxylated quartz surfaces are dominated by vicinal and germinal silanols (Muster et al. (2001) and references therein) which then tend to create network of H-bonds with each other (Musso et al., 2011; 2012), therefore, not really free and "dangling" per se.

We have changed this to refer to surface OH groups.

**Technical comments**

Since the manuscript exclusively talks about ice nucleating ability of various minerals, I would suggest the term "material/materials" be replaced by "mineral/minerals" throughout the manuscript (eg. line 19, 171, 207 etc.) The text has been altered accordingly.

Line 24-28 This is a long sentence. This can be split into 2 to better convey the meaning, eg. 2nd sentence could be "the ice nucleation particle population in desert

dust aerosol is dominated by Kfeldspars rather than quartz (or other minerals)." Thank you. The abstract has been edited accordingly.

Line 41 "type of aerosol particles" instead of "aerosol types" Now corrected

Line 43 "Field observations of ice crystal residuals" instead of "Observations of aerosol at the centre of ice crystals" We acknowledge this suggestion but have edited the text as follows. We wanted to avoid the term residual, since those unfamiliar with the field won't know what it is. The other referee also had a suggestion for this sentence. "Observations of aerosol within ice crystals have shown that mineral dust is often present…"

Line 44 Cite the accepted ACP version of Eriksen Hammer et al. (2018) Now updated.

Line 46-49 May consider splitting this into "Atmospheric mineral dusts are composed of several components. Clay is the major component of airborne mineral dust and is sufficiently small that its atmospheric lifetime is relatively long. Hence recent ice nucleation studies have focused on the clay group of minerals." Thank you, this has now been reworded as follows "Atmospheric mineral dusts are composed of several components. Clay is the major component of airborne mineral dust and is sufficiently small that its atmospheric lifetime is relatively long. Hence, historically ice nucleation studies have focused on the clay group of minerals (Broadley et al., 2012;Murray et al., 2011;Wex et al., 2014;Mason and Maybank, 1958;Pinti et al., 2012;Roberts and Hallett, 1968)."

Should be 'INPs' in place of 'ice nucleants' (line 51) and 'ice nucleating material' (line 71) corrected

Line 67 'repeated' instead of 'repeat' Corrected

Line 70 remove 'quite' Amended

Line 74 "over time when suspended in water or exposed to air" would be better in place of 'to time spent in water or air'. Thank you for this suggestion. However, the particles were not suspended in water during the time series experiments. They were left in water over time and so would settle to the base of the vial. They would only be suspended prior to the start of a new experiment. We have amended the text as follows "We also explore the stability of a subset of these samples to time spent in water or exposed to air"

Line 104 TiO2 is mentioned twice. Remove one of them Corrected

Line 139 this line gives the impression that the samples undergoing BET tests were used to make suspensions, which I believe is not the case. Consider removing "After BET analysis" In some instances we had small amounts of sample so that we did have to use the same sample as what was used in the BET tests. We have added the following section of text as clarification "In some instances we had small amounts of sample so the sample used for BET analysis was subsequently used for the succeeding ice-nucleation experiments."

Line 176-177 delete 'specific' from line 176 and add it before "surface area" in line 177 Thank you, this is now amended

Line 293-295 give references after 'configuration'

This is our own inference based on our understanding of the system.

Line 332 "It may be these microtextural differences that lead to the observed variability in icenucleating ability" Amended

Line 377-379 Consider re-phrasing the lines to "However, we constrained the polynomial fits because the unconstrained fits poorly represented the data at the warmest and coldest ends." for easy understanding. We have reworded the text as follows "These fits were constrained at the warmest and coldest temperatures due to a poor representation of the data at these regimes when left unconstrained."

Line 415 "its exceptionally high ice nucleation ability" in place of 'it exhibiting exceptional behaviour' This has been rephrased as follows "Amelia albite from the Harrison et al. (2016) study was excluded due to its exceptional ice-nucleating ability making it unrepresentative of the other five albite samples."

Line 447 can also add Kumar et al. (2018); Kumar et al. (2019b) in the references in regards to aging of K-feldspar microcline Thank you. The references have been added.

Line 471-472 Fig 8b shows parameterisation for desert dust by Niemand et al. (2012) in blue dashed lines and the K-feldspar parameterisation proposed in this study by solid red lines Amended as follows "K-feldspar parameterisation developed by Atkinson et al. (2013) (in black dashed lines), the parameterisation for desert dust by Niemand et al. (2012) (blue dashed lines) and the K-feldspar parameterisation proposed here (red solid lines)."

Lines 506-510 Weathering in solutes has already been addressed by Whale et al. (2018) and 3 Parts series from Kumar et al. and should be addressed as part of this paper rather than referring to future work We have now added the following section of text "Related to this, we also note that solutes can alter the ice-nucleating ability of mineral samples (Whale et al., 2018;Kumar et al., 2018;Kumar et al., 2019a, b). Sensitivity to these ageing processes and solutes could be very important in determining the dominant INP types globally (Boose et al., 2019). Hence, we suggest further studies aim to build a better understanding of the relationship between the experimental observations and field collected samples to determine the role of ageing in the atmosphere."

Line 642, 666 and 673 - cite the accepted ACP version of these papers: Kumar et al. (2019a), Peckhaus et al. (2016), Pinti et al. (2012) Amended

Figure 2 add suspension concentration in caption Done

Figure 5 Niedermeier et al. (2015) - correct author name in figure legend. Also, the curves are difficult to read, especially with orange-red tones, maybe use other dark colors. Size of symbols in legend needs to be increased for better readability Done

Figure 8 Check references to Boose et al 2016 (for 'a' and/or 'b') for data used in the figure. Symbol size in legend should be increased. Corrected

In line 826 "….the natural mineral variability of freshly milled quartz highlighted…." Line 795-796 equation formatting needs improvement Corrected

We would also like to bring to the attention of the reviewers that an erratum has been issued for Zolles et al. (2015). As a result the quartz parameterisation from this study has been amended. We have also added quartz data from a volcanic ash study to increase the data density for quartz (Losey et al., 2018). This has resulted in a slightly modified milled quartz parameterisation which is then used in the proceeding plots and discussions. The amended parameterisation did not significantly change and it has had no impact on the discussion and conclusions of the paper.

**References**

[revised manuscript text omitted]

---

## Editor Comment (EC1) · Hinrich Grothe (Editor) · 21 Jul 2019

I may add theses completed references:

Boose, Y., Baloh, P., Plötze, M., Ofner, J., Grothe, H., Sierau,B., Lohmann, U., Kanji, Z.A.: Heterogeneous ice nucleation on dust particles sourced from nine deserts world-wide – Part 2: Deposition nucleation and condensation freezing, Atmos. Chem. Phys., 19, 1059–1076, 2019, 10.5194/acp-19-1059-2019

Zolles, T., Burkart, J., Häusler, T., Pummer, B., Hitzenberger, R., Grothe, H.: Correction to "Identification of Ice Nucleation Active Sites on Feldspar Dust Particles" J. Phys.

[Figure]

Chem. A 2019, 10.1021/acs.jpca.9b05645

Interactive
comment